# Efficient Coding of Natural Images using Maximum Manifold Capacity Representations

**Thomas Yerxa** [1]   **Yilun Kuang** [2,3]   **Eero Simoncelli** [1,2,3]   **SueYeon Chung**[1,2]

[1]Center for Neural Science, New York University
[2]Center for Computational Neuroscience, Flatiron Institute
[3]Courant Inst. of Mathematical Sciences,
`tey214@nyu.edu`

## Abstract

The efficient coding hypothesis posits that sensory systems are adapted to the statistics of their inputs, maximizing mutual information between environmental signals and their representations, subject to biological constraints. While elegant, information theoretic quantities are notoriously difficult to measure or optimize, and most research on the hypothesis employs approximations, bounds, or substitutes (e.g., reconstruction error). A recently developed measure of coding efficiency, the "manifold capacity", quantifies the number of object categories that may be represented in a linearly separable fashion, but its calculation relies on a computationally intensive iterative procedure that precludes its use as an objective. Here, we simplify this measure to a form that facilitates direct optimization, use it to learn Maximum Manifold Capacity Representations (MMCRs), and demonstrate that these are competitive with state-of-the-art results on current self-supervised learning (SSL) recognition benchmarks. Empirical analyses reveal important differences between MMCRs and the representations learned by other SSL frameworks, and suggest a mechanism by which manifold compression gives rise to class separability. Finally, we evaluate a set of SSL methods on a suite of neural predictivity benchmarks, and find MMCRs are highly competitive as models of the primate ventral stream.

## 1   Introduction

Biological visual systems learn complex representations of the world that support a wide range of cognitive behaviors, without relying on a large number of labelled examples. The efficient coding hypothesis (8; 65) suggests that this is accomplished by adapting the sensory representation to the statistics of the input signal, so as to reduce redundancy or dimensionality. Visual signals have several clear sources of redundancy. They evolve slowly in time, since temporally adjacent inputs typically correspond to different views of the same scene, which in turn are usually more similar than views of distinct scenes. Moreover, the variations within individual scenes often correspond to variations in a small number of parameters, such as those controlling viewing and lighting conditions, and are thus inherently low dimensional. Many previous results have demonstrated how the computations of neural circuits can be seen as matched to such structures in naturalistic environments (48; 29; 65; 17) Studies in various modalities have identified geometric structures in neural data that are associated with behavioral tasks (10; 25; 43; 33; 56), and explored metrics for quantifying these structures.

The recent development of "manifold capacity theory" provides a more explicit connection between the geometry (size and dimensionality) of neural representations and their coding capacity (18). This theory has been used to evaluate efficiency of biological and artificial neural networks across modalities (17; 24; 66; 52). However, usage as a design principle for building model representations has not been explored.

37th Conference on Neural Information Processing Systems (NeurIPS 2023).

Motivated by these observations, we seek to learn representations in which manifolds containing different views of the same scene are both compact and low-dimensional, while manifolds corresponding to distinct scene are maximally separated. Specifically:

- We develop a form of Manifold Capacity that can be used as an objective function for learning.

- We demonstrate that a Maximum Manifold Capacity Representation (MMCR) supports high-quality object recognition (matching the SoTA for self-supervised learning), when evaluated using the standard linear evaluation paradigm (i.e., applying an optimized linear classifier to the output of the self-supervised network) (14).

- Through a analysis of internal representations and learning signals, we analyze the underlying mechanism responsible for the emergence of semantically relevant features from unsupervised objective functions.

- We validate the effectiveness of MMCR as a brain model by comparing its learned representations against neural data obtained from Macaque visual cortex.

Our work thus leverages normative goals of representational efficiency to obtain a novel model for visual representation that is both effective for recognition and consistent with neural responses in the primate ventral stream.

## 1.1 Related Work

**Geometry of Neural Representations.** Previous work has sought to characterize how representational geometry, often measured through spectral quantities like the participation ratio (the squared ratio of the $l_1$ and $l_2$ norms of the eigenvalues of the covariance matrix), shapes different aspects of performance on downstream tasks (32). Elmoznino and Bonner (27) found that high dimensionality in ANN representations was associated with ability to both predict neural data and generalize to unseen categories. Stringer et al. (67) observed that the spectrum of the representation of natural images in mouse cortex follows a power law with a decay coefficient near 1, and Agrawal et al. (2) report that (in artifical representations) the proximity of the spectral decay coefficient to one is an effective predictor of how well a representation will generalize to downstream tasks.

**Self-Supervised Learning.** Our methodology is related to (and inspired by) recent advances in contrastive self-supervised representation learning (SSL), but has a distinctly different motivation and formulation. Many recent frameworks craft objectives that are designed to maximize the mutual information between representations of different views of the same object (58; 14; 58; 68; 4)). However, estimating mutual information in high dimensional feature spaces is difficult (9), and furthermore it is not clear that closer approximation of mutual information in the objective yields improved representations (71) By contrast, capacity measures developed in spin glass theory (35; 1) are derived in the "large N (thermodynamic) limit"and thus are intended to operate in the regime of large ambient dimension (18; 5). We examine whether one such measure, which until now had been used only to evaluate representation quality, is also useful as an objective function for SSL.

Many SSL methods minimize the distance between representations of different augmented views of the same image while employing constraints to prevent collapse to trivial solutions (e.g., repulsion of negative pairs (14), or feature space whitening (77; 7; 28)). The limitations of using a single pairwise distance comparison have been demonstrated, notably in the development of the "multi-crop" strategy implemented in SwAV (13) and in the contrastive multiview coding approach Tian et al. (68). Our approach is based on the assumption that different views of an image form a continuous manifold that we aim to compress. We characterize each set of image views with the spectrum of singular values of their representations, using the nuclear norm as a combined measure of the manifold size and dimensionality.

The nuclear norm has been previously used to infer or induce low rank structure in the representation of data (42; 72; 49). In particular, Wang et al. (72) use it as a regularizer to supplement an InfoNCE loss. Our approach represents a more radical departure from the traditional InfoNCE loss, as we detail below. Rather than pair a low-rank prior with a logistic regression-based likelihood, we make the more symmetric choice of employing a *high rank* likelihood. This allows the objective to explicitly discourage dimensional collapse, a well known issue in SSL (45).

## 2    Maximum manifold capacity representations

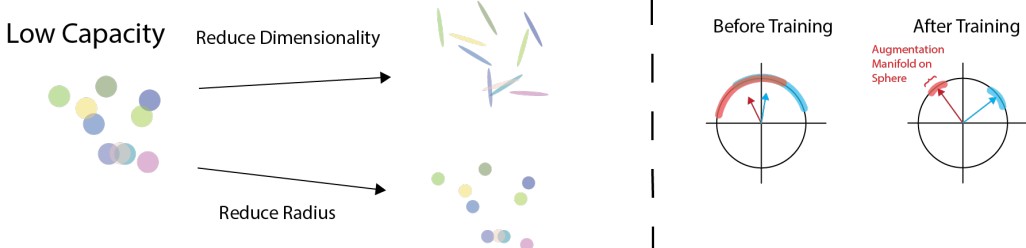

Figure 1: Two dimensional illustrations of high and low capacity representations. Left: the capacity (linear separability) of a random set of spherical regions can be improved, either by reducing their radii (while maintaining their dimensionalities), or by reducing their dimensionalities (while maintaining their average radii). Right: the objective proposed in this paper aims to minimize the nuclear norm (equal to the product of radius and sqrt dimensionality) of normalized data vectors (ie., lying on the unit sphere). Before training, the manifolds have a large extent and thus the matrix of their corresponding centroid vectors has low nuclear norm. After training, the capacity is increased: The manifolds are compressed and repelled from each other, resulting in centroid matrix with larger nuclear norm and lower similarity.

### 2.1    Manifold Capacity Theory

Consider a set of $P$ manifolds embedded in a feature space of dimensionality $D$, each assigned a class label. Manifold capacity theory is concerned with the question: what is the largest value of $\frac{P}{D}$ such that there exists (with high probability) a hyperplane separating a random dichotomy (22; 34)? Recent theoretical work has demonstrated that there exists a critical value, dubbed the manifold capacity $\alpha_C$, such that when $\frac{P}{D} < \alpha_C$ the probability of finding a separating hyperplane is approximately 1.0, and when $\frac{P}{D} > \alpha_C$ the probability is approximately 0.0 (18). The capacity $\alpha_C$ can be accurately predicted from three key quantities: (1) the manifold radius $R_M$, which measures the size of the manifold relative to the distance of its centroid from the origin, (2) the manifold dimensionality $D_M$ which quanitifies the number of dimensions along which a manifold has significant extent, and (3) the correlation of the manifold centroids. When the centroid correlation is low the manifold capacity can be approximated by $\phi(R_M\sqrt{D_M})$ where $\phi(\cdot)$ is a monotonically decreasing function.

For manifolds of arbitrary geometry, the radius and dimensionality may be computed using an iterative process that alternates between determining the set of "anchor points" on each manifold that are relevant for the classification problem, and computing the statistics of random projections of these anchor points (20). This process is both computationally costly and non-differentiable, and therefore unsuitable for use as an objective function. For more detail on the general theory see Appendix C. However, if the manifolds are assumed to be elliptical in shape, then both radius and dimensionality may be expressed analytically:

$$R_M = \sqrt{\sum_i \lambda_i^2}, \qquad D_M = \frac{(\sum_i \lambda_i)^2}{\sum_i \lambda_i^2}, \tag{1}$$

where the $\lambda_i^2$ are the eigenvalues of the covariance matrix of the manifold. For comparison, when computing these values for a set of 100 128-D manifolds with 100 points sampled from each, the use of the analytical expression is approximately 500 times faster (in "wall-clock time") than the general iterative procedure.

Using these definitions for manifold radius and dimensionality we can write the capacity as $\alpha_C \approx \phi(\sum_i \sigma_i)$ where $\sigma_i$ are the singular values of a matrix containing points on the manifold (equivalently, the square roots of the eigenvalues of the covariance matrix). In this form, the sum is the $L_1$ norm of the singular values, known as the *Nuclear Norm* of the matrix. When used as an objective function, this measure favors sparse solutions (i.e., those with a small number of non-zero singular values) corresponding to low dimensionality. It is worth comparing this objective to another natural candidate for quantifying size: the determinant of the covariance matrix. The determinant is equal to the product

of the eigenvalues (which captures the squared volume of the corresponding ellipsoid), but lacks the preference for lower dimensionality that comes with the Nuclear Norm. Specifically, since the determinant is zero whenever one (or more) eigenvalue is zero, it cannot distinguish zero-volume manifolds of different dimensionality. Lossy coding rate (entropy) has also been used as a measure of compactness (76), which simplifies to the log determinant of the covariance matrix under a Gaussian model (50). In this case, the identity matrix is added to a multiple of the feature covariance matrix before evaluating the determinant, which solves the dimensionality issue described above.

## 2.2 Optimizing Manifold Capacity

Now we construct an SSL objective function based on manifold capacity. For each input image (notated as a vector $\mathbf{x}_b \in \mathbb{R}^{\mathbb{D}}$) we generate $k$ samples from the corresponding manifold by applying a set of random augmentations (each drawn from the same distribution), yielding manifold sample matrix $\tilde{\boldsymbol{X}}_b \in \mathbb{R}^{D \times k}$. Each augmented image is transformed by a Deep Neural Network, which computes nonlinear function $f(\mathbf{x}_b; \theta)$ parameterized by $\theta$, and the $d$-dimensional responses are projected onto the unit sphere yielding manifold response matrix $\boldsymbol{Z}_b \in \mathbb{R}^{d \times k}$. The centroid $\boldsymbol{c}_b$ is approximated by averaging across the columns (response vectors). For a set of images $\{\mathbf{x}_1, ..., \mathbf{x}_B\}$ we compute normalized response matrices $\{\boldsymbol{Z}_1, ..., \boldsymbol{Z}_B\}$ and assemble their corresponding centroids into matrix $\boldsymbol{C} \in \mathbb{R}^{d \times B}$.

Given the responses and their centroids, the MMCR loss function can be written simply:

$$\mathcal{L} = -||\boldsymbol{C}||_*, \tag{2}$$

where $|| \cdot ||_*$ indicates the nuclear norm. The loss explicitly maximize the extent of the "centroid manifold", which interestingly is sufficient to learn a useful representation. Concretely, maximizing $||\boldsymbol{C}||_*$ implicitly minimizes the extent of each individual object manifold as measured by $||\boldsymbol{Z}_b||_*$. We build intuition for this effect below.

**Compression by Maximizing Centroid Nuclear Norm Alone.** Each centroid vector is a mean of unit vectors, and thus has a norm that is linearly related to the average cosine similarity of those unit vectors. Specifically,

$$||\boldsymbol{c}_b||^2 = \frac{1}{K} + \frac{2}{K^2} \sum_{k=1}^{K} \sum_{l=1}^{k-1} \boldsymbol{z}_{b,k}^T \boldsymbol{z}_{b,l} \tag{3}$$

Here $\boldsymbol{z}_{b,i}$ denotes the representation of the $i^{th}$ augmentation of $\boldsymbol{x}_b$. We can gain further insight by considering how the distribution of singular vectors of a matrix depends on the norms and pairwise similarities of the constituent column vectors. While no closed form solution exists for the singular values of an arbitrary matrix, the case where the matrix is composed of two column vectors can provide useful intuition. If $\boldsymbol{C} = [\boldsymbol{c}_1, \boldsymbol{c}_2]$, $\boldsymbol{Z}_1 = [\boldsymbol{z}_{1,1}, \boldsymbol{z}_{1,2}]$, $\boldsymbol{Z}_2 = [\boldsymbol{z}_{2,1}, \boldsymbol{z}_{2,2}]$, the singular values of $\boldsymbol{C}$ and $\boldsymbol{Z}_i$ are:

$$\boldsymbol{\sigma}(\boldsymbol{C}) = \sqrt{\frac{||\boldsymbol{c}_1||^2 + ||\boldsymbol{c}_2||^2 \pm ((||\boldsymbol{c}_1||^2 - ||\boldsymbol{c}_2||^2)^2 + 4(\boldsymbol{c}_1^T \boldsymbol{c}_2)^2)^{1/2}}{2}}, \tag{4}$$

$$\boldsymbol{\sigma}(\boldsymbol{Z}_i) = \sqrt{1 \pm \boldsymbol{z}_{i,1}^T \boldsymbol{z}_{i,2}}. \tag{5}$$

So, $||\boldsymbol{\sigma}(\boldsymbol{C})||_1 = ||\boldsymbol{C}||_*$ is maximized when the centroid vectors have maximal norms (bounded above by 1, since they are the centroids of unit vectors), and are orthogonal to each other. As we saw above the centroid norm is a linear function of within-manifold similarity. Similarly, $||\boldsymbol{\sigma}(\boldsymbol{Z}_i)||_1 = ||\boldsymbol{Z}_i||_*$ is minimized when the within-manifold similarity is maximal. Thus the single term $||\boldsymbol{C}||_*$ encapsulates both of the key ingredients of a contrastive learning framework, and we will demonstrate below that simply maximizing $||\boldsymbol{C}||_*$ is sufficient to learn a useful representation. This is because the compressive role of "positives" in contrastive learning is carried out by forming the centroid vectors, so the objective is not positive-free. For example, if only a single view is used the objective lacks a compressive component and fails to produce a useful representation. In Appendix F we demonstrate empirically that this implicit form effectively reduces $||\boldsymbol{Z}_b||_*$ by comparing to the case where $||\boldsymbol{Z}_b||_*$ is minimized explicitly. So, all three factors which determine the manifold capacity (radius,

dimensionality, and centroid correlations) can be succinctly expressed in an objective function with a single term, $-||\boldsymbol{C}||_*$.

**Computational Complexity.** Evaluating the loss for our method involves computing a singular value decomposition of $\boldsymbol{C} \in \mathbb{R}^{d \times B}$ which has complexity $\mathcal{O}(Bd \times \min(B, d))$, where $B$ is the batch size and $d$ is the dimensionality of the output. By comparison, contrastive methods that compute all pairwise distances in a batch have complexity $\mathcal{O}(B^2 d)$ and non-contrastive methods that involve regularizing the covariance structure have complexity $\mathcal{O}(Bd^2)$. Additionally, the complexity of our method is constant with respect to the number of views used (though the feature extraction phase is linear in the number of views), while pairwise similarity metrics have complexity that is quadratic in the number of views. It is also worth noting that doing implicit compression by maximizing $||\boldsymbol{C}||_*$ offers an advantage in computational complexity relative to explicit compression. This is because evaluating a term such as $\sum_{b=1}^{B} ||\boldsymbol{Z}_b||_*$ has computational complexity of $\mathcal{O}(B^2 d \times \min(B, d))$.

### 2.3 Conditions for Optimal Embeddings

Recently HaoChen et al. (39) developed a framework based on spectral decomposition of the "population augmentation graph", which provides theoretical guarantees for the performance of self-supervised learning on downstream tasks under linear probing. This work was extended to provide insights into various other SSL objectives by Balestriero and LeCun (6), and we show below that leveraging this approach can lead to explicit conditions for the optimality of representation under our proposed objective as well.

Given a dataset $\boldsymbol{X}' = [\boldsymbol{x}_1, ..., \boldsymbol{x}_N]^T \in \mathbb{R}^{N \times D'}$ we construct a new dataset by creating $k$ randomly augmented views of the original data, $\boldsymbol{X} = [\text{view}_1(\boldsymbol{X}'), ..., \text{view}_k(\boldsymbol{X}')] \in \mathbb{R}^{Nk \times D}$. The advantage of doing so is that we can now leverage the knowledge that different views of the same underlying datapoint are *semantically related*. We can express this notion of similarity in the symmetric matrix $\boldsymbol{G} \in \{0, 1\}^{Nk \times Nk}$ with $\boldsymbol{G}_{ij} = 1$ if augmented datapoints $i$ and $j$ are semantically related (and $\boldsymbol{G}_{ii} = 1$ as any datapoint is related to itself). We can normalize $\boldsymbol{G}$ such that its rows and columns sum to 1 (so rows of $\boldsymbol{G}$ are $k$-sparse with nonzero entries equal to $1/k$).

Now let $\boldsymbol{Z} \in \mathbb{R}^{Nk \times d}$ be an embedding of the augmented dataset. Then we have $\boldsymbol{GZ} = [\boldsymbol{C}, ..., \boldsymbol{C}]^T$ where $\boldsymbol{C}$ is the matrix of centroid vectors introduced above, and the number of repetitions of $\boldsymbol{C}$ is $k$. Then because $\boldsymbol{\sigma}([\boldsymbol{C}, ..., \boldsymbol{C}]) = \sqrt{k}\boldsymbol{\sigma}(\boldsymbol{C})$ we can write MMCR loss function as,

$$\mathcal{L} = -||\boldsymbol{GZ}||_* \tag{6}$$

This connection allows us to make the following statements about the optimal embeddings $\boldsymbol{Z}$ under our loss, which we prove in Appendix A:

**Theorem:** Under the proposed loss, the left singular vectors of an optimal embedding, $\boldsymbol{Z}^*$, are the eigenvectors of $\boldsymbol{G}$, and the singular values of $\boldsymbol{Z}^*$ are proportional to the top $d$ eigenvalues of $\boldsymbol{G}$.

## 3 Results

**Architecture.** For all experiments we use ResNet-50 (40) as a backbone architecture (for variants trained on CIFAR we removed max pooling layers). Following Chen et al. (14), we append a small perceptron to the output of the average pooling layer of the ResNet so that $z_i = g(h(x_i))$, where $h$ is the ResNet and $g$ is the MLP. For ImageNet-1k/100 we used an MLP with dimensions $[8192, 8192, 512]$ and for smaller datasets we used $[512, 128]$.

**Optimization.** We employ the set of augmentations proposed in (38). For ImageNet we used the LARS optimizer with a learning rate of 4.8, linear warmup during the first 10 epochs and cosine decay thereafter with a batchsize of 2048, and pre-train for 100 epochs. Note that though we report results using a default batch size of 2048, a batch size as small as 256 can be used to obtain reasonable results (1.2% reduction compared to batch size 2048 – see Appendix J for a sweep over batch size). We additionally employ a momentum encoder for ImageNet pre-training (all views are fed through an online network and an additional network whose parameters are a slowly moving average of the online network and all embeddings of each image are averaged to form centroids). We found the use of a momentum encoder provided a small advantage in terms of downstream classification

Table 1: Evaluation of learned features on downstream classification tasks. The leftmost columns show results for the standard frozen-linear evaluation procedure on ImageNet (IN). Results for most methods in this setting are taken from Ozsoy et al. (59) except for SwAV which is taken from the original paper (13). Columns 2 and 3 show semi-supervised evaluation on ImageNet (fine-tuning on 1% and 10% of labels). The results for this setting for VICReg and CorInfo Max are copied from Ozsoy et al. (59). The final three columns show frozen-linear evaluation on other datasets. We evaluated models for which pretrained weights in the 100 epoch setting were available online; MoCo, Barlow Twins and BYOL were taken from solo-learn da Costa et al. (23) (`https://github.com/vturrisi/solo-learn`), while SwAV and SimCLR were taken from VISSL Goyal et al. (37) (`https://github.com/facebookresearch/vissl/blob/main/MODEL_ZOO.md`). For all evaluations we performed we report the mean and standard deviation over 3 evaluation runs).

| Method | IN | 1% | 10% | Food-101 | Flowers-102 | DTD |
|---|---|---|---|---|---|---|
| W-MSE (28) | 69.4 | - | - | - | - | - |
| NNCLR (26) | 69.4 | - | - | - | - | - |
| SwAV (13) | 64.6 | - | - | - | - | - |
| SimSiam (15) | 68.1 | - | - | - | - | - |
| CorInfoMax (59) | 69.1 | 44.9 | 64.3 | - | - | - |
| VICReg (7) | 68.7 | 44.8 | 62.2 | - | - | - |
| BarlowTwins (77) | 68.7 | $45.1 \pm .12$ | $61.7 \pm .03$ | $69.8 \pm .03$ | $86.2 \pm .22$ | $67.7 \pm .20$ |
| SimCLR (14) | 66.5 | $42.6 \pm .02$ | $61.6 \pm .09$ | $67.2 \pm .24$ | $84.0 \pm .19$ | $64.8 \pm .07$ |
| BYOL (38) | 69.3 | $49.8 \pm .05$ | $65.0 \pm .05$ | $70.6 \pm .1$ | $84.8 \pm .43$ | $67.6 \pm .14$ |
| MoCo-V2 (16) | 67.4 | $43.4 \pm .07$ | $63.2 \pm .07$ | $68.6 \pm .03$ | $82.4 \pm .27$ | $66.6 \pm .16$ |
| SwAV (13) | **72.1** | $49.8 \pm .09$ | $66.9 \pm .05$ | $72.1 \pm .08$ | $89.3 \pm .1$ | $68.2 \pm .18$ |
| MMCR (2 views) | $69.5 \pm .02$ | $46.6 \pm .02$ | $63.9 \pm .02$ | $72.0 \pm .02$ | $90.0 \pm .24$ | $68.5 \pm .07$ |
| MMCR (4 views) | $71.5 \pm .04$ | $49.4 \pm .05$ | $66.0 \pm .05$ | $73.2 \pm .07$ | $91.0 \pm .04$ | **70.4** $\pm .46$ |
| MMCR (8 views) | **72.1** $\pm .04$ | **51.0** $\pm .02$ | **67.7** $\pm .11$ | **73.6** $\pm .04$ | **91.4** $\pm .07$ | $70.0 \pm .24$ |

performance, see Appendix O for this ablation. For smaller CIFAR-10 we used a smaller batch size, many more views (40), and the Adam optimizer with fixed learning rate. See Appendix D for exact details.

## 3.1 Transfer to Downstream Tasks

We used a standard linear evaluation technique, freezing the parameters of the encoding network and training a linear classifier with supervision (14), to verify that our method extracts semantically relevant features from the data. We also perform semi-supervised evaluation, where all model parameters are fine tuned using a small number of labelled examples, and also check whether the learned features generalize to three out-of-distribution datasets: Flowers-102, Food-101, and the Describable Textures Dataset (57; 11; 19) . The results are summarized in Table 1 and details of the training of the linear classifier are provided in Appendix G. Finally we also evaluate our best model in and each of the baselines on object detection on the VOC07 dataset. We followed (41; 77), fine tuning the representation networks for detection with a Faster R-CNN head and C-4 backbone using the 1x schedule. MMCR achieved a mean average precision (mAP) of 54.6 and the baseline method performance ranged from 53.1 to 56.0, demonstrating that though our framework was inspired by a theory of classification the learned features do generalize to other vision tasks. See Appendix M for detailed results.

Note that all included models were trained using the same backbone architecture (ResNet-50), dataset (ImageNet-1k), and number of pretraining epochs (100; we briefly explore the impact of longer pretraining in Appendix P ). The results for networks pretrained on smaller datasets can be found in the Appendix I.

## 3.2 Analyses of Learned Representations

We next conduct a series of experiment to elucidate the differences between representations learned with different SSL procedures, and suggest a mechanism by which augmentation manifold compres-

sion gives rise to class separability. To reduce the computational requirements, these analyses are carried out on models trained on the CIFAR-10 dataset.

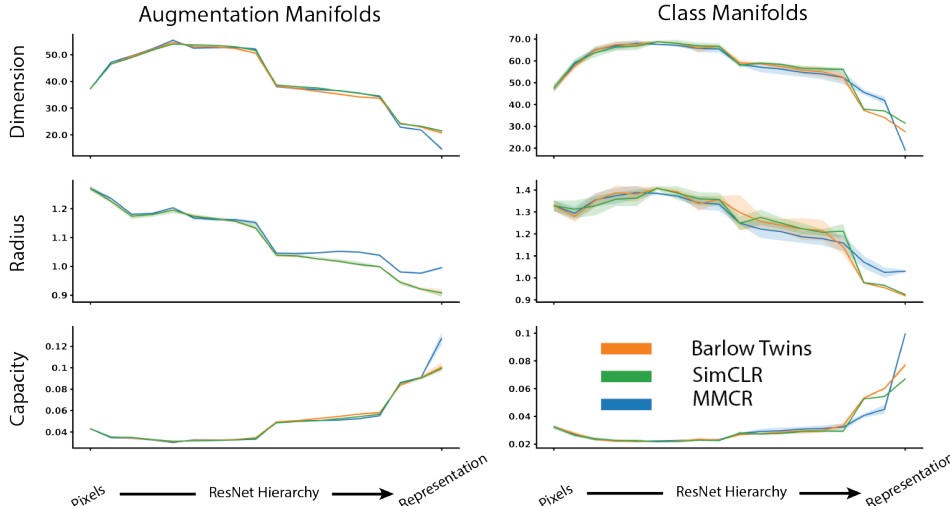

Figure 2: Mean field Manifold Capacity analysis. Manifold dimensionality (top row), radius (middle row), and capacity (bottom row), as a function of stage in the representational hierarchy (from pixel inputs to output of the encoder/learned representation). Shaded regions indicate 95% confidence intervals around the mean (analysis was conducted with 5 different random samples from the dataset, see Appendix E).

**Mean Field Theory Manifold Capacity.** In Fig. 2 we show that our representation, which is optimized using an objective that assumes elliptical manifold geometry, nevertheless yields representations with high values of the more general mean field manifold capacity (relative to baseline methods). For completeness we also analyzed the geometries of class manifolds, whose points are the representations of different examples from the same class. This analysis provided further evidence that learning to maximize augmentation manifold capacity compresses and separates class manifolds, leading to a useful representation. Interestingly MMCRs seem to use a different strategy than the baseline methods to increase the capacity, yieldingclass/augmentation manifolds with larger radii, but lower dimensionality (Fig. 2) These geometrical differences do not emerge until the tail end of the hierarchy, suggesting early layers of each network are carrying out stereotyped transformations and loss function induced specialization does not emerge until later layers.

**Emergence of neural manifolds via gradient coherence.** We hypothesize that class separability in MMCRs arises because augmentation manifolds corresponding to examples from the same class are optimally compressed by more similar transformations than those stemming from distinct classes. To investigate this empirically, we evaluate the gradient of the objective function for inputs belonging to the same class. We can then check whether gradients obtained from (distinct) batches of the same class are more similar to each other than those obtained from different classes, which would suggest that the strategy for compressing augmentation manifolds from the same class are relatively similar to each other. Fig. 3 demonstrates that this is the case: within class gradient coherence, as measured by cosine similarity, is consistently higher than across class coherence across both training epochs and model hierarchy.

**Manifold subspace alignment.** Within-class gradient coherence constitutes a plausible mechanistic explanation for the emergence of class separability, but it does not explain why members of the same class share similar compression strategies. To explore this question we examine the geometric properties of augmentation manifolds in the pixel domain. Here we observe small but measurable differences between the distributions of within-class similarity and across-class similarity, as demonstrated in the top row of Fig. 4. The subtle difference in the geometric properties of augmentation manifolds in the pixel domain in turn leads to the increased gradient coherence observed above, which leads to a representation that rearranges and reshapes augmentation manifolds from the same class in a similar fashion (bottom row of Fig. 4), thus allowing better linear separation of classes. Not only are centroids

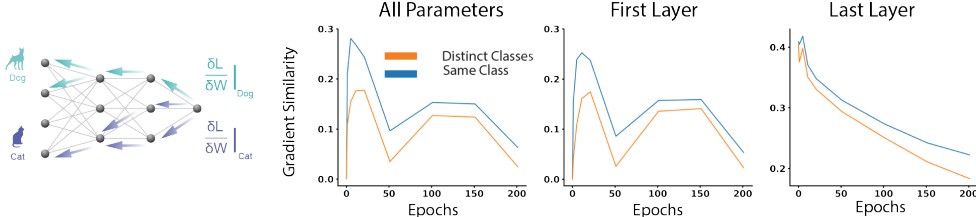

Figure 3: Cosine similarity of gradients for pairs of single-class batches. We plot the mean pairwise similarity for pairs of gradients for different subsets of the model parameters (all parameters, and the first and last linear operators) obtained from single-class-batches coming from the same or distinct classes over the course of training. Because a large number of bathces were used 95% confidence intervals about these means are too small to be visible. To the left is a visualization of the fact that single-class gradients flow backward through the model in more similar directions.

of same-class-manifolds in more similar regions of the representation space than those coming from distinct classes (Fig. 4 third column bottom row) but additionally same-class-manifolds have more similar shapes to each other (Fig. 4 bottom row columns 1 and 2 show same-class-manifolds occupy subspaces with lower relative angles and share more variance).

We next ask how the representation learned with the MMCR objective differs from those optimized for other self-supervised loss functions. While MMCR encourages centroids to be orthogonal, the InfoNCE loss (14) encourages negative pairs to be as dissimilar as possible, which is achieved when they lie in opposite regions of the *same* subspace . The Barlow Twins (77) loss is not an explicit function of feature vector similarities, but instead encourages individual features to be correlated and distinct features to be uncorrelated, across the batch dimension. Fig. 5 shows that these intuitions are borne out empirically: the MMCR representation produces augmentation manifold centroids that are significantly more orthogonal to each other than the two baseline methods.

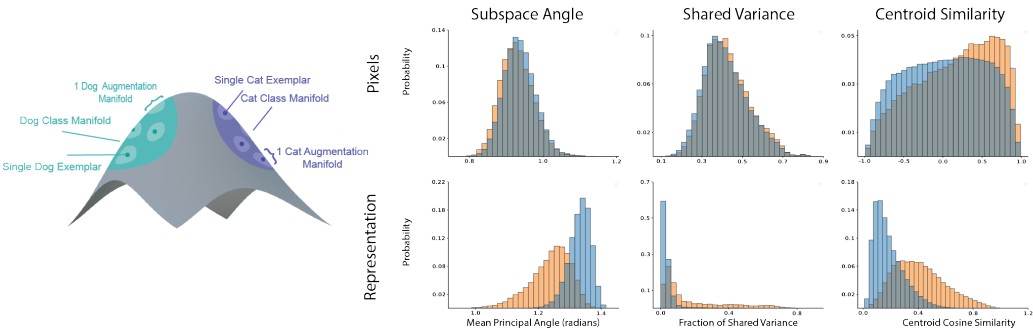

Figure 4: The distributions of various similarity metrics for augmentation manifolds from the same (orange) and distinct (blue) classes. These are shown for both the input images (top row), and the learned representation (bottom row). Left: schematic illustration of the exemplar-augmentation manifold-class manifold structure of the learned representation.

### 3.3 Biological relevance

Neuroscience has provided motivation for many of the developments in artifical neural networks, and it is of interest to ask whether SSL networks can characterize the measured behaviors of neurons in biological visual systems. As a simple test, Table 2 shows performance of our model compared with five other SSL models on the *BrainScore* repository (64; 30; 53; 51). We find that MMCR achieves the highest performance in explaining neural data from primate visual areas V2 and V4, second-highest for V1 and IT, and is the most or second most predictive for 8 out of the 11 individual datasets (see Appendix K).

In addition, we examine general response spectral characteristics that have been recently described for neural populations. In particular, Stringer et al. (67) reported that the eigenspectrum of the

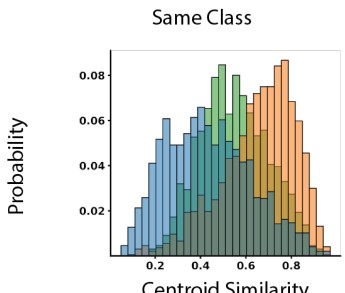
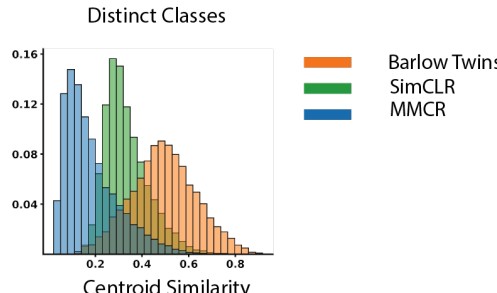

Figure 5: Cosine similarities of centroids for models trained according to different SSL objectives. The left panel shows the distribution of cosine similarities for centroids of augmentation manifolds for examples of the same class, while the right shows the same distribution for examples from distinct classes. Note that because we are analyzing the outputs of the ResNet backbone (which are rectified), the minimum possible cosine similarity is $0$.

Table 2: Neural prediction and spectral properties of self-supervised models. First four columns provide average BrainScore correlation values (64) for neurons recorded in four areas of primate ventral stream. Last two provide participation ratio (PR) and spectral decay coefficients ($\alpha$), estimated for 10 bootstrapped samples of the features for the ImageNet validation set. When estimating the spectral decay coefficient we omitted the tails (where the power decays more rapidly), see L for exact details. Entries indicate mean and standard error of the mean, and boldface indicates best performance within standard error. MMCR results are for a model trained with 8 views.

| Model | V1 | V2 | V4 | IT | PR | $\alpha$ |
|-------|-----|-----|-----|-----|-----|-----|
| MMCR | **.494** $\pm$ .006 | **.311** $\pm$ .005 | **.481** $\pm$ .005 | .416 $\pm$ .003 | 279.2 $\pm$ .3 | 1.04 $\pm$ 1e-4 |
| SimCLR | **.500** $\pm$ .007 | .288 $\pm$ .007 | .475 $\pm$ .004 | **.420** $\pm$ .003 | 124.6 $\pm$ .1 | 1.34 $\pm$ 3e-4 |
| BYOL | **.500** $\pm$ .006 | .291 $\pm$ .007 | **.477** $\pm$ .005 | .404 $\pm$ .003 | 248.0 $\pm$ .4 | 1.35 $\pm$ 1e-4 |
| MoCo | **.499** $\pm$ .007 | .293 $\pm$ .006 | **.477** $\pm$ .005 | **.417** $\pm$ .003 | 147.7 $\pm$ .2 | 1.44 $\pm$ 3e-4 |
| Barlow | **.498** $\pm$ .007 | .293 $\pm$ .008 | **.477** $\pm$ .005 | .404 $\pm$ .003 | 252.2 $\pm$ .2 | 1.21 $\pm$ 3e-4 |
| SwAV | .488 $\pm$ .007 | .296 $\pm$ .009 | .463 $\pm$ .004 | .398 $\pm$ .003 | 163.9 $\pm$ .2 | 1.15 $\pm$ 3e-4 |

covariance matrix of population activity in visual area V1 follows a power law decay with a decay coefficient of approximately $1$ ($\lambda_n \propto n^{-\alpha}$ with $\alpha \approx 1$ where $\lambda_n$ is the $n^{th}$ eigenvalues). Subsequent studies in artifical networks have found that such a decay spectrum is associated with increased robustness to adversarial perturbations and favorable generalization properties (55; 2). Additionally, several recent works have investigated the connection between representational dimensionality and neural predictivity (70; 63). In particular, Elmoznino and Bonner (27) report that high intrinsic dimensionality (as measured with the participation ratio of the representation covariance) is correlated with stronger ability to predict neural activity. Table 2 provides values for the participation ratio of each representation over the ImageNet validation set, as well as the decay coefficient of the covariance spectrum (see Appendix L for more details on each experiment). We see that the MMCR has the highest participation ratio (dimensionality) - note that this differs from the quantity optimized in the objective function, which lies in the embedding space. In addition, MMCR also also yields features with a decay coefficient that is nearest to one.

We find that in this controlled setting each model explains a very similar fraction of neural variance through linear regression. This is consistent with recent and concurrent works (21; 62) which have identified this lack of ability to discriminate between alternative models as a weakness of the dominant paradigm used for model-to-brain comparisons. However, our results demonstrate that different SSL algorithms produce representations with meaningfully different geometries (as evidenced by the large spread in the spectral properties such as the participation ratio and decay coefficient). This suggests the need for the development of new metrics for comparing models to data, such as geometrical measures, that capture these important differences between candidate models.

# 4 Discussion

We have presented a novel self-supervised learning algorithm inspired by manifold capacity theory. Many existing SSL methods can be categorized as either "contrastive" or "non-contrastive" depending on whether they avoid collapse by imposing constraints on the embedding gram or covariance matrix, respectively. Our framework strikes a compromise, optimizing the singular values of the embedding matrix itself. By directly optimizing this population level feature (the spectrum), we are able to encourage alignment and uniformity (71) simultaneously with a single-term objective. Additionally, this formulation circumvents the need for making a large number of pairwise comparisons, either between instances or dimensions. As a result learning MMCRs is efficient, requiring neither large batch size nor large embedding dimension. Finally, our method extends naturally to the multi-view case, offering improved performance with minimal increases in computational cost.

Our formulation approximates manifold geometries as elliptical, reducing computational requirements while still yielding a useful learning signal and networks with high manifold capacity. Specifically, we were able to leverage manifold capacity analysis in its full generality to gain insight into the geometry of MMCR networks after training. Further research could explore objectives based on more modest reductions of mean field manifold capacity that capture non-elliptical structure. Intriguingly, our method produces augmentation and class manifolds with lower dimensionality but larger radius than either Barlow Twins or SimCLR (Fig. 2). We do not understand why this is the case, but the differences indicate that capacity analysis can provide a useful tool for elucidating the different encoding strategies encouraged by various SSL paradigms.

Finally we investigated two recently proposed theories on ideal spectral properties for neural representations. For our considered set of models a spectral decay coefficient near 1 was associated with better performance on the within-distribution task and generalization to unseen datasets (MMCR, SwAV, and Barlow were the top three models for each of the out-of-distribution classification tasks), a finding which is broadly aligned with both the empirical and theoretical findings of Agrawal et al. (2). However, we also found that high dimensionality did not always correspond to strong neural predictivity: Despite having the lowest dimensionality, SimCLR performed strongly in terms of neural predictivity. This implies, perhaps unsurprisingly, that global dimensionality alone is not sufficient to explain the response properties of neurons in the primate ventral stream. In fact our experiments add to a growing body of work on the need for complementary approaches to linear predictivity for discriminating between candidate models of the visual system (21). Happily the field is already moving to address this limitation, for instance concurrent work (12) finds that decomposing neural predictivity error into distinct modes can yield insights into how different models fit different aspects of the neural data (even if they yield similar overall predictivity).

One promising direction for improving the quality of artificial networks as models of neural computations is to incorporate the constraints associated with biologically plausible learning (i.e. the need for local learning rules, Illing et al. (44)). A complementary direction is to better align training data diets with ecological inputs. For example, temporal invariance rather than augmentation invariance seems a more plausible objective for a neural system to optimize (73; 3; 61). We speculate that a variant of the MMCR objective that operates over time may be well-suited to a neural circuit implementation as its computation would only require mechanisms for tracking (1) short timescale temporal means (to form centroids) and (2) the singular values of the population activity over longer timescales.

Neuroscience is brimming with newly collected datasets recorded from ever larger populations of neurons, and there is a long history of methods that aim to make sense of these measurements through a normative lens. Here we have demonstrated that one such technique that has proven useful for gleaning insights from neural data (60; 75; 31) can be reformulated for use as an objective function to learn a useful abstract representation of images. Future work should aim to close the loop between modelling and analysis by using these learned models to generate experimentally testable predictions and constrain new experimental designs.

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

## A  Optimal Embeddings

Recall the setting of self-supervised learning as described in Balestriero and LeCun (6): given a dataset $\boldsymbol{X'} = [\boldsymbol{x}_1, ..., \boldsymbol{x}_N]^T \in \mathbb{R}^{N \times D'}$ we construct a new dataset by creating $k$ randomly augmented views of the original data, $\boldsymbol{X} = [\text{view}_1(\boldsymbol{X'}), ..., \text{view}_k(\boldsymbol{X'})] \in \mathbb{R}^{Nk \times D}$. The advantage of doing so is that we can now leverage the knowledge that different views of the same underlying datapoint are *semantically related*. We can express this notion of similarity in the symmetric matrix $\boldsymbol{G} \in \{0, 1\}^{Nk \times Nk}$ with $\boldsymbol{G}_{ij} = 1$ if augmented datapoints $i$ and $j$ are semantically related (and $\boldsymbol{G}_{ii} = 1$ as any datapoint is related to itself). We can normalize $\boldsymbol{G}$ such that its rows and columns sum to 1 (so rows of $\boldsymbol{G}$ are $k$-sparse with nonzero entries equal to $1/k$).

Now let $\boldsymbol{Z} \in \mathbb{R}^{Nk \times d}$ be an embedding of the augmented dataset. Then we have $\boldsymbol{GZ} = [\boldsymbol{C}, ..., \boldsymbol{C}]^T$ where $\boldsymbol{C}$ is the matrix of centroid vectors introduced above, and the number of repetitions of $\boldsymbol{C}$ is $k$. Then because $\boldsymbol{\sigma}([\boldsymbol{C}, ..., \boldsymbol{C}]) = \sqrt{k}\boldsymbol{\sigma}(\boldsymbol{C})$ we can write MMCR loss function as,

$$
\begin{aligned}
\mathcal{L} &= -||\boldsymbol{GZ}||_* \\
&= -||\boldsymbol{Q\Lambda Q}^T \boldsymbol{USV}^T||_* \\
&= -||\boldsymbol{\Lambda Q}^T \boldsymbol{US}||_*
\end{aligned}
\tag{7}
$$

Where we have taken the eigendecomposition of $\boldsymbol{G}$ which is real and symmetric and the SVD of $\boldsymbol{Z}$, and then used the fact that the singular value spectrum is invariant under left or right orthogonal transformations. We now show that a global optima of this objective is achieved when the left singular vectors of $\boldsymbol{Z}$ are the eigenvectors of $\boldsymbol{G}$ and the singular values of $\boldsymbol{Z}$ are proportional to the eigenvalues of $\boldsymbol{G}$. Throughout we will assume that the size of the dataset is greater than the dimensionality of the embedddings, $N > d$, as is the case in practical applications. First we prove a simple lemma about the spectrum of matrices who are extended by zeros (i.e. embedded in a higher dimensional space).

**Lemma A.1**: For $\boldsymbol{A} \in \mathbb{R}^{N \times N}$, $\boldsymbol{B} \in \mathbb{R}^{N \times d}$ with $d < N$, $||\boldsymbol{AB}||_* = ||\boldsymbol{A\tilde{B}}||_*$ where $\tilde{\boldsymbol{B}} = [\boldsymbol{B}, \boldsymbol{0}] \in \mathbb{R}^{N \times N}$.

**Proof**: First note that $\boldsymbol{A\tilde{B}} = [\boldsymbol{AB}, \boldsymbol{0}]$ so it suffices to show that for arbitrary $\boldsymbol{X}$ that $\boldsymbol{\sigma}(X) = \boldsymbol{\sigma}([\boldsymbol{X}, \boldsymbol{0}])$. Taking the SVD of $\boldsymbol{X}$,

$$
\boldsymbol{X} = \begin{bmatrix} \boldsymbol{U} & \widetilde{\boldsymbol{U}} \end{bmatrix} \begin{bmatrix} \boldsymbol{\Sigma} \\ \boldsymbol{0} \end{bmatrix} \begin{bmatrix} \boldsymbol{V}^T \end{bmatrix} = \boldsymbol{U\Sigma V}^T
$$

Then a valid singular value decomposition for $\tilde{X}$ is

$$
\tilde{\boldsymbol{X}} = \begin{bmatrix} \boldsymbol{U} & \widetilde{\boldsymbol{U}} \end{bmatrix} \begin{bmatrix} \boldsymbol{\Sigma} & \boldsymbol{0} \\ \boldsymbol{0} & \boldsymbol{0} \end{bmatrix} \begin{bmatrix} \boldsymbol{V}^T & \boldsymbol{0} \\ \boldsymbol{0} & \boldsymbol{I} \end{bmatrix}
$$

Clearly then, $||\boldsymbol{X}||_* = ||\tilde{\boldsymbol{X}}||_*$

**Theorem:** The proposed loss achieves a global minimum when the left singular vectors of $\boldsymbol{Z}$ are the eigenvectors of $\boldsymbol{G}$, and the singular values of $\boldsymbol{Z}$ are proportional to the top $d$ eigenvalues of $\boldsymbol{G}$.

**Proof:** Let $\tilde{\boldsymbol{Z}} = [\boldsymbol{Z}, \boldsymbol{0}] \in \mathbb{R}^{N \times N}$. By Lemma A.1 we have $||\boldsymbol{GZ}||_* = ||\boldsymbol{G\tilde{Z}}||_*$. Von Neumann's trace inequality can be used to show $||\boldsymbol{G\tilde{Z}}||_* \leq \sum_{i=1}^{Nk} \boldsymbol{\sigma}_i(\boldsymbol{G})\boldsymbol{\sigma}_i(\tilde{\boldsymbol{Z}})$ (see Marshall et al. (54) for proof). Examining (4) it is clear that this bound is achieved when $\boldsymbol{U} = \boldsymbol{Q}$. The problem can therefore be reduced to the constrained optimization problem,

$$
\min_{\boldsymbol{\sigma}_i(\tilde{\boldsymbol{Z}})} \sum_{i=1}^{Nk} \boldsymbol{\sigma}_i(\boldsymbol{G})\boldsymbol{\sigma}_i(\tilde{\boldsymbol{Z}})
$$

$$
\text{subject to } \sum_{i=1}^{Nk} \boldsymbol{\sigma}_i(\tilde{\boldsymbol{Z}})^2 = Nk
$$

where the constraint comes from the fact that columns of $\boldsymbol{Z}$ are unit vectors. Intuitively, we are maximizing the inner product between a fixed vector $\boldsymbol{\sigma}(\boldsymbol{G})$ and a vector with fixed L2 norm. The

solution of course is to align the two vectors as closely as possible, i.e. when $\boldsymbol{\sigma}_i(\tilde{\boldsymbol{Z}}) \propto \boldsymbol{\sigma}_i(\boldsymbol{G})$ for $i = 1, ..., d$. It is worth noting that by construction $\boldsymbol{\sigma}_i(\tilde{\boldsymbol{Z}}) = 0$ for $i > d$ and the columns of $\boldsymbol{U}$ associated with these zero valued singular values are unconstrained.

## B  Pytorch Style Pseudocode for MMCR

```
1  # h: encoder
2  # g: projection head
3  # T: momentum temperature
4  # B: batch size
5  # K: number of augmentations
6  # D: projector output dimensionality
7  #
8  # lmbda: trade-off parameter
9
10 f_o, g_o = ResNet50(), MLP() # online networks
11
12 # initialize momentum network with identical params
13 f_m, g_m = f_o.copy, g_o.copy()
14
15 # momentum networks are not updated via gradient descent
16 f_m.requires_grad = False
17 g_m.requires_grad = False
18
19 for x in loader:
20     # K randomly augmented views
21     x = multi_augment(x) # B x K x H x W
22
23     # push through encoder and projector
24     z_o = g_o(h_o(x)) # B x K x D
25     z_m = g_m(h_m(x)) # B x K x D
26     z = concatenate(z_o, z_m, dim=1) # append outputs
27
28     # project onto unit sphere
29     z = normalize(z, dim=-1)
30
31     # calculate centroids (mean over augmentation axis)
32     c = z.mean(dim=1) # B x D
33
34     # calculate singular values
35     U_z, S_z, V_z = svd(z) # batch svd
36     U_c, S_c, V_c = svd(c)
37
38     # calculate loss
39     loss = -1.0 * sum(S_c) + lmbda * sum(S_z) / B
40
41     # backward pass and optimization step
42     loss.backward()
43     optim.step()
44
45     # perform momentum update
46     with torch.no_grad():
47         f_m.parameters() = (1 - T) * f_o.parameters() + T * f_m.
    parameters()
48         g_m.parameters() = (1 - T) * g_o.parameters() + T * g_m.
    parameters()
```

## C  Mean Field Theory Manifold Capacity Background Information

For completeness we summarize some of the central arguments from Chung et al. (18), which develops the general form of manifold capactiy theory.

**Mean Field Theory** Recall the problem setting for manifold capacity analysis: given a set of $P$ manifolds embedded in a feature space of dimensionality $D$, each assigned a random binary class label (18). Manifold capacity theory is concerned with the question: what is the largest value of $\frac{P}{D}$ such that there exists (with high probability) a hyperplane separating the two classes? In the thermodynamic limit, where $P, D \to \infty$ but $\frac{P}{D}$ remains finite, the inverse capacity can be written exactly,

$$\alpha_M^{-1} = \mathbb{E}_{\vec{T}}[F(\vec{T})] \tag{8}$$

where, $F(\vec{T}) = \min_{\vec{V}} \left\{ \|\vec{V} - \vec{T}\|^2 \mid g_{\mathcal{S}}(\vec{V}) \geq 0 \right\}$, $\mathcal{S}$ is the set defining the manifold geometry (i.e. the set of vectors $\vec{S}$ that are points on an individual manifold), $\vec{T}$ are random vectors drawn from a white multivariate Gaussian distribution, and $g_{\mathcal{S}}(\vec{V}) = \min_{\vec{S}}\{\vec{V} \cdot \vec{S} \mid \vec{S} \in \mathcal{S}\}$, is the concave support function.

The KKT equations for this convex optimization problem are:

$$
\begin{aligned}
\vec{V} - \vec{T} - \lambda \tilde{S}(\vec{T}) &= 0 \\
\lambda &\geq 0 \\
g_{\mathcal{S}}(\vec{V}) - \kappa &\geq 0 \\
\lambda \left[ g_{\mathcal{S}}(\vec{V}) - \kappa \right] &= 0.
\end{aligned} \tag{9}
$$

, where $\tilde{S}(\vec{T})$ is a subgradient of the support function. When the support function is differentiable, the subgradient is unique and equal to the gradient,

$$\tilde{S}(\vec{T}) = \nabla g_{\mathcal{S}}(\vec{V}) = \arg\min_{\vec{S} \in \mathcal{S}} \vec{V} \cdot \vec{S} \tag{10}$$

$\tilde{S}(\vec{T})$ is the unique point in the convex hull of $\mathcal{S}$ that satisfies the first KKT equation, and is called the "anchor point" for $\mathcal{S}$ induced by the random vector $\vec{T}$.

**Equivalent Interpretation of Anchor Points** For a given dichotomy (random binary class labelling) the weight vector of the maximum margin separating hyperplane can be decomposed into a sum of at most $P$ vectors, with each manifold contributing a single vector, which lies within the convex hull of the manifold. The position of said point point is a function of the manifolds position relative to all of the other manifolds in the space and depends on the particular set of random labels. Thus there exists a distribution of separating-hyperplane-determining-points for each individual manifold. Using the cavity method it can be shown that these points are none other than the anchor points that are involved in solving the optimization problem described above (36).

**Numerical Solution** To solve the mean field equations numerically, one samples several random Gaussian vectors $\vec{T}$, and then for each $\vec{T}$, $\vec{V}$ and $\vec{S}$ are determined by solving the quadratic programming program given above. The capacity is then estimated as the mean value of $F$ or the samples $\vec{T}$.

**Manifold Geometries** The way the capacity varies in terms of the statistics of the anchor points can be simplified by introducing two key quantities, the manifold radius $R_M$ and manifold dimensionality $R_M$:

$$
\begin{aligned}
R_M^2 &= \mathbb{E}_{\vec{T}}[\|\tilde{S}(\vec{T})\|^2] \\
D_M &= \mathbb{E}_{\vec{T}}[\vec{T} \cdot \hat{S}(\vec{T})]
\end{aligned} \tag{11}
$$

where $\hat{S}(\vec{T})$ is a unit-vector in the direction of the anchor point $\tilde{S}$. In particular as discussed in the main text, the manifold capacity can be approximated by $\phi(R_M\sqrt{D_M})$ where $\phi$ is a monotonically decreasing function.

**Elliptical Geometries** In the case where the manifolds exhibit elliptical symmetries, the manifold radius and dimensionality can be written in terms of the eigenvalues of the covariance matrix of the anchor points:

$$R_M^2 = \sum_i \lambda_i^2$$
$$D_M = \frac{\left(\sum_i \lambda_i\right)^2}{\sum_i \lambda_i^2}$$

(12)

So, in this case $R_M$ is the total variability of the anchor points, and $D_M$ is a generalized participation ratio of the anchor point covariance, a well known soft measure of dimensionality.

## D   Additional Pre-training information

**Settings for CIFAR/STL-10** We take the parameters of each augmentation directly from Zbontar et al. (77), but for these lower resolution images we omitted Gaussian blurring and solarization augmentations. All models were trained for 500 epochs using the Adam optimizer (46) with a learning rate of $1e-3$ and weight decay of $1e-6$. For all three methods we used a one hidden layer MLP with hidden dimension of 512 and output dimension of 128 for the projector head $g$. We swept batch size for each method and chose the one that resulted in the highest downstream task performance. For both SimCLR and Barlow Twins we found that a batch size of 128 was optimal (among 32, 64, 128, 256, and 512) for all 3 datasets. For MMCR there is a trade-off between batch size and the number of augmentations used, and the optimal value of that trade-off is highly dataset dependent. For CIFAR-10 and CIFAR-100 we used batch size of 32 and 40 views, and for STL-10 we used a batch of 64 with 20 views For Barlow Twins we used $\lambda = \frac{1}{128}$ which normalizes for the number of elements in the on-diagonal and off-diagonal terms in the loss. For SimCLR we used the recommended setting of $\tau = 0.5$. The overall performance of both baseline methods (and likely MMCR as well) could be increased with a more thorough hyperparameter search and by employing methodology that more closely matches the original works. For example, both methods would likely benefit from the combination of larger batch size, the use of the LARS optimizer (which is designed for large batch optimization), a learning rate scheduler consisting of linear warm-up followed by cosine annealing, longer training, and the use of more diverse augmentations (i.e. including solarization and gaussian blur). Additionally Barlow Twins reports that the representation can benefit from using a much larger projector network than we use. Because our goal was primarily to demonstrate that MMCR can produce representations that are comparable to these baselines rather than to produce state-of-the-art results on small scale datasets we opted for simplifications wherever possible (using off the shelf Adam for optimization with a fixed learning rate, and fixing architectural hyperparameters like the projector dimensionality).

**Settings for ImageNet-100** For ImageNet we more closely match the pre-training procedures of previous works. We use a batch size of 2048 and a smaller number of views for MMCR (4), and also use the full suite of augmentations from Zbontar et al. (77). For the sake of efficiency we train for a reduced number of epochs (200). For MMCR and SimCLR we modified the projector hidden dimensionality to be 4096 for the projector head, following the original work (14). For Barlow Twins we used the recommended 2-layer MLP with hidden and output dimensions of 8192, and set $\lambda = 5e-3$, however these hyperparameters were optimal for the full ImageNet dataset, and not neccesarrily for ImageNet-100. We were unable to achieve better downstream performance using a ResNet-50 backbone than what has previously been reported in the literature for this dataset with a ResNet-18 backbone, therefore we report the ResNet-18 performance reported in (23). For SimCLR we use $\tau = 0.1$ which is the recommended setting for larger batch sizes.

**Settings for ImageNet-1k**: For ImageNet-1k we use mostly identical settings to ImageNet-100, but we increased the capacity of the projector network (using a 2 hidden layer MLP with hidden dimenisons of 8192 and output dimension of 512). We scaled the learning rate linearly with batch size: lr $= 0.6 \times \frac{\text{batch size}}{256}$. Additionally we reduce the number of pretraining epochs to 100. Finally for ImageNet-1k we found that employing a momentum encoder slightly boosted downstream performance (around +1% on ImageNet frozen-linear evaluation). Specifically, an identical encoding network and projector architecture is initialized with the same parameters as the initial "online" network, and during training the weights of this "momentum" network track a slowly moving average of the online network parameters (only the online network parameters are updated via gradient

descent). Each augmented view is passed through both the online and momentum networks, and the resultant embeddings are all averaged to form the centroid vector for a particular image in the batch. We used a momentum coefficient of 0.99.

Pre-training on 16 A100 GPUs using 8 views (our most compute intensive setting) takes approximately 32 hours.

# E    Details of Representational Analyses

## E.1    Manifold Capacity Analysis

For each pre-trained model, we extract layer activations across the ResNet hierarchy after a forward pass of a set of images. For class manifold analysis, the set of images contain 10 classes, where each class has 100 examples. Augmentation manifolds instead have 100 exemplars with 100 examples each. Following (20), we take activations from all convolutional layers in ResNet-50 after a ReLU non-linearity. The specific extracted layers highlighted in bold fonts are given by Table 3. The final analysis results are averaged over five data samplings with different random seeds and random projections of intermediate features to lower-dimension spaces (default 5000 dimensions).

## E.2    Gradient Coherence Analysis

In Fig. 3, for each of the classes of CIFAR-10, we generate 100 batches of 32 augmentation manifolds of samples from a specific class (with 40 augmentations each). We then measure the gradient of the loss function for each batch during different stages of training, and compute the cosine similarity between every pair of gradients. Across all stages of training the mean cosine similarity between gradients generated from batches of the same class is larger than those from distinct classes (left column). This observation remains true when isolating the gradients of parameters from different stages of in the resnet-50 hierarchy (center and right columns, respectively).

## E.3    Manifold Subspace Alignment

For Fig. 4 we generated 100 samples from the augmentation manfiolds of 500 images in the CIFAR-10 dataset. We then measure the mean subspsace angle (left column), fraction of shared variance (middle column) and centroid cosine similarity between each pair of manifolds. The same procedure was used for generating the data for Fig. 5.

**Subspace Angle.** Besides measuring the size and dimensionality of individual object manifolds we also wish to characterize the degree of overlap between pairs of manifolds. For this, we measure the angle between their subspaces (47), which is a generalization of the notion of angles that applies to subspaces of arbitrary dimension.

**Shared Variance.** Object manifolds will generally have a lower intrinsic dimensionality then the space in which they are embedded. Therefore, the data will have low variance along several of the principal vectors used to calculate the set of subspace angles, and so many of the principal angles will have little meaning. To address this limitation we also compute the shared variance between the linear subspaces that contain object manifolds.

# F    Implicit MMCR Effectively Reduces Augmentation Manifold Nuclear Norm

To test whether or not implicit manifold compression actually reduces the mean augmentation manifold nuclear norm, we can vary the value of $\lambda$. Below we see the evolution of both terms of the loss for several different values of lambda during training on CIFAR-10. For these experiments the batch size was 64 and the number of augmentations per image was 4.0. As shown in Fig. 6, the level of compression of individual manifolds is nearly the same across all values of the parameter.

Table 3: A Total of 18 Extracted ResNet-50 Layers (in **Bold**) for MFTMA Analysis

| Layer | Type | Conv2d Size (H × W × C) |
|---|---|---|
| pixel | **Input** | None |
| conv1 | $\begin{bmatrix} \text{Conv2d} \\ \text{BatchNorm} \\ \textbf{ReLU} \end{bmatrix} \times 1$ | $[7 \times 7 \times 64] \times 1$ |
| conv2_x | $\begin{bmatrix} \begin{bmatrix} \text{Conv2d} \\ \text{BatchNorm} \\ \textbf{ReLU} \end{bmatrix} \times 3 \end{bmatrix} \times 3$ | $\begin{bmatrix} 1 \times 1 \times 64 \\ 3 \times 3 \times 64 \\ 1 \times 1 \times 256 \end{bmatrix} \times 3$ |
| conv3_x | $\begin{bmatrix} \begin{bmatrix} \text{Conv2d} \\ \text{BatchNorm} \\ \textbf{ReLU} \end{bmatrix} \times 3 \end{bmatrix} \times 4$ | $\begin{bmatrix} 1 \times 1 \times 128 \\ 3 \times 3 \times 128 \\ 1 \times 1 \times 512 \end{bmatrix} \times 4$ |
| conv4_x | $\begin{bmatrix} \begin{bmatrix} \text{Conv2d} \\ \text{BatchNorm} \\ \textbf{ReLU} \end{bmatrix} \times 3 \end{bmatrix} \times 6$ | $\begin{bmatrix} 1 \times 1 \times 256 \\ 3 \times 3 \times 256 \\ 1 \times 1 \times 1024 \end{bmatrix} \times 6$ |
| conv5_x | $\begin{bmatrix} \begin{bmatrix} \text{Conv2d} \\ \text{BatchNorm} \\ \textbf{ReLU} \end{bmatrix} \times 3 \end{bmatrix} \times 3$ | $\begin{bmatrix} 1 \times 1 \times 512 \\ 3 \times 3 \times 512 \\ 1 \times 1 \times 2048 \end{bmatrix} \times 3$ |

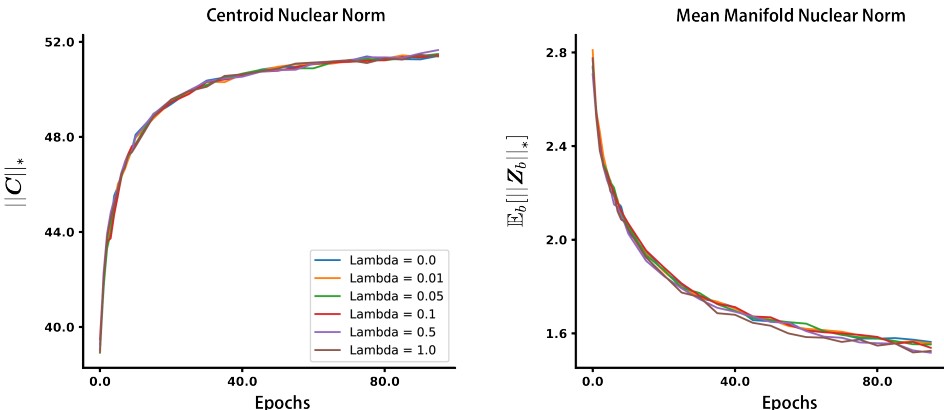

Figure 6: Validation loss values for different values of $\lambda$

## G   Classification Evaluation Procedure

**CIFAR and STL-10**: During pre-training all models were monitored with a k-nearest neighbor classifier (k=200) and checkpointed every 5 epochs. After pre-training, we trained linear classifiers on all checkpoints whose monitor accuracy was within 1% of the highest observed accuracy, and select the model that achieves the highest linear classification accuracy. Linear classifiers were trained using the Adam optimizer with batch size of 1024 and an initial learning rate of 0.1, which decayed according to a cosine scheduler over the course of 50 epochs. For the linear classifier training, at train time we use the same set of augmentations as during unsupervised pretraining, at test time we only use center cropping and random horizontal flipping.

**ImageNet-1k/100**: For ImageNet datasets we closely followed the most widely adopted evaluation procedure. Following pre-training we freeze the encoder weights and train a linear layer in a supervised fashion using SGD with a batch size of 2048, learning rate of 1.6, and weight decay of 1e-6 for 50 epochs. During linear classifier training the only data augmentations are random cropping and random horizontal flips, and during evaluation inputs are center cropped.

**Semi-Supervised**: For semi-supervised evaluation we mostly follow the procedure outlined in Bardes et al. (7). We use the SGD optimizer with momentum of 0.9 and weight decay of 1e-6 and the standard cross entropy loss. The augmentation procedure was the same as described above. Because the linear classifier is being trained from scratch and the representation is being fine tuned the learning rate for the parameters of the backbone is scaled down by a factor of 10, and both learning rates (backbone and classifier) followed a cosine decay schedule for 20 epochs. We used a batch size of 256 and swept the learning rate over [0.1, 0.3, 1.0] for each model.

**Other Downstream Classification Tasks**: Classifiers on these datasets were trained in a similar fashion to those trained on the CIFAR and STL-10 datasets. The only differences being that for each method and datasets we swept the batch size over [128, 256, 512, 1024] and the the initial learning rate over [3e-2, 3e-3, 3e-4], and the augmentation procedure matched the standard setup for ImageNet training (only random cropping and horizontal flipping for training, resizing and center cropping for evaluation).

## H  Training Metrics

In the Fig. 7 below we monitor the evolution of both the objective (second panel), the mean augmentation manifold nuclear norm, the centroid norm, and the mean centroid similarity evaluated on the test set over the course of training.

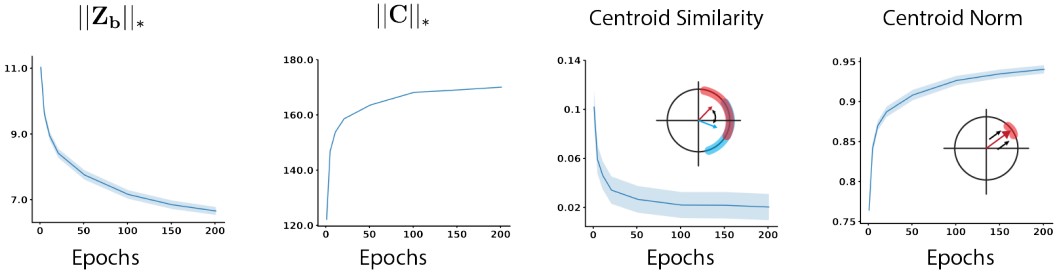

Figure 7: Evolution of various metrics during training. Geometric measures are evaluated on a set of 200 manifolds, each defined by an image drawn from the CIFAR-10 dataset, along with 16 augmentations. Shaded regions indicate a 95% confidence interval around the mean.

## I  Classification Performance on Smaller Datasets

In Table 4 below we report the performance of both our method as well as Barlow Twins and SimCLR when trained using a ResNet-50 backbone on smaller datasets.

## J  Batch Size Dependence

One of the most cited drawbacks of contrastive SSL methods has been that strong performance on downstream tasks requires training with large batch sizes, while non-contrastive methods (e.g., VICReg or Barlow Twins (77; 7)) that place constraints on the cross-correlation/covariance matrices of the embeddings are much more amenable to smaller batch training. It is also worth noting that the need for large batch sizes in contrastive methods can be alleviated in various ways, such as maintaining a memory bank (74) or employing a slowly updating momentum encoder (41). Given that our method is neither wholly contrastive nor non-contrastive (since it acts on the spectrum of the embedding matrix directly), we wondered howwould depend on training batch size. We pretrained on ImageNet-1k using batch sizes of $256, 512, 1024, 2048, 4096$ and evaluate the linear classification

Table 4: Top-1 classification accuracies of linear classifiers for representations trained with various datasets and objective functions. Note: for Barlow Twins on ImageNet-100 we report the result from da Costa et al. (23) which uses a ResNet-18 backbone, as we were unable to obtain better performance. For MMCR on ImageNet-100 we tested both 2 views (matched to baselines) and 4 views, results are formatted (2-view)/(4-view)

| Method | CIFAR-10 | CIFAR-100 | STL-10 | ImageNet-100 |
|---|---|---|---|---|
| Barlow Twins (our repro.) | 90.91 | 67.91 | 89.96 | 80.38* |
| SimCLR (our repro.) | 92.22 | 70.04 | **91.11** | 79.64 |
| MMCR ($\lambda = 0.0$) | **93.53** | 69.87 | 90.62 | 81.52/**82.88** |
| MMCR ($\lambda = 0.01$) | 93.39 | **70.94** | 90.77 | 81.28/82.56 |

accuracy for each. Encouragingly we observed only a modest decrease in performance for the smallest batch size tested. The results of this sweep, in comparison to Barlow Twins and SimCLR, is shown in in Fig. 8 Note that for these runs we used two views and the linear learning rate scaling as described in Appendix D. Future work should endeavor to better understand the impact of various hyperparameters on the quality of learned representations.

An important detail is that for this experiment we did not employ a momentum encoder when training MMCR. It is argued in He et al. (41) that the momentum encoder increases the effective minibatch size as the slowly moving weights encode information from preceding batches. However here we are interested explicitly interested in batch size dependence so we ablate this architectural confound (neither Zbontar et al. (77); Chen et al. (14) employ momentum encoders).

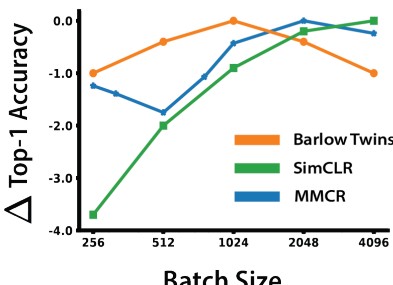

Figure 8: Drop in top-1 performance relative to that of the best setting for three methods. Data for both Barlow Twins and SimCLR are copied from Zbontar et al. (77).

## K  Additional Details on BrainScore

Brain-Score evaluates a model in terms of its ability to predict the measured responses of neurons to images. We provide a brief introduction to the metric here (see (64) for a complete description). Let $y \in \mathbb{R}^N$ denote the average response of a single (biological) neuron to a set of $N$ training images and $X \in \mathbb{R}^{N \times K}$ be the response of $K$ model neurons to the same images. Brain Score first solves the linear regression problem $y = Xw$ for weights $w$. The model then predicts responses $y'$ to a set of held out images. Next the Pearson correlation coefficient between $y'$ and y is calculated, and the score for a particular dataset is the median of the individual neuron predicitivities in said dataset. The mean of these scores is taken over different train-test splits of each dataset. The score for a total brain area (as shown in Table 2 is the mean over train-test splits and distinct datasets. The standard error of the means for each dataset (which are typically between 1e-3 and 1e-2) within an area are summed quadrature and divided by the number of datasets to produce the errors reported in table 2. In table 5 we split the brain area scores into individual datasets.

| Model | V1.0 | V1.1 | V2.0 | V4.0 | V4.1 | V4.2 |
|---|---|---|---|---|---|---|
| MMCR | 0.270 | 0.718 | 0.311 | 0.577 | 0.627 | 0.492 |
| SimCLR | 0.224 | 0.776 | 0.288 | 0.576 | 0.626 | 0.48 |
| BYOL | 0.274 | 0.727 | 0.291 | 0.585 | 0.626 | 0.48 |
| MoCo | 0.273 | .726 | 0.293 | 0.57 | 0.629 | 0.492 |
| Barlow | 0.276 | .721 | 0.293 | 0.568 | 0.626 | 0.493 |
| SwAV | 0.252 | .723 | 0.296 | 0.568 | 0.614 | 0.469 |

| Model | V4.3 | IT.0 | IT.1 | IT.2 | IT.3 | |
|---|---|---|---|---|---|---|
| MMCR | 0.226 | 0.554 | 0.558 | 0.545 | 0.424 | |
| SimCLR | 0.224 | 0.552 | 0.545 | 0.518 | 0.456 | |
| BYOL | 0.216 | 0.55 | 0.545 | 0.516 | 0.41 | |
| MoCo | 0.215 | 0.54 | 0.560 | 0.550 | 0.437 | |
| Barlow | 0.221 | 0.545 | 0.547 | 0.518 | 0.412 | |
| SwAV | 0.202 | 0.533 | 0.537 | 0.518 | 0.405 | |

Table 5: Brain-score comparison of six self-supervised models, over 11 different electrophysiological data sets recorded from macaque monkeys (64). Datasets V1.0 and V2.0 are from Freeman et al. (30), V1.1 is from Marques et al. (53), and V4.0, V4.1, IT.0, and IT.1 are from Majaj et al. (51). (V4/IT).(2/3) are the SanghaviJozwik2020 and SanghaviMurty2020 datasets as denoted by brainscore.

## L  Additional Details on Spectral Properties

**Participation Ratio** We first extracted the 2048 dimensional feature vectors for each model in response to the images in the ImageNet validation set. Images were resized to $256 \times 256$ and then center cropped to $224 \times 224$ following the setting in which the classifiers are tested. We resampled the resultant feature matrices with replacement 10 times independently for each model For each resampled dataset (of features) we calculate the empirical covariance matrix, associated eigenspectra, and associated participation ratios (squared ratio of $L_1$ to $L_2$ norm of the eigenvcetors).

**Decay Coefficient** The spectra obtained using the procedure outlined above all decayed rapidly near the tails (the least significant eigenvalues). To avoid undo bias from these tails when estimating the decay coefficients we only considered the top 2000 eigenvalues. To estimate the decay coefficient we fit a regression line to the logarithm of the eigenvalues as a function of the logarithm of their indexes. This fitting procedure was repeated across the bootstrapped spectra for each model to obtain standard errors of the mean. For all models the linear regression produced a strong fit to the data, with the minimum observed $R^2$ value being $0.95$.

## M  Object Detection

To ensure that the representations obtained with MMCR are not hyperspecialized to classification tasks we also evaluated our highest performing model on object detection. We follow (41; 77), fine tuning the representation network with a Faster R-CNN head and C-4 backbone on the VOC07+12 dataset (training with the train+val split and evaluating on the VOC07 test split). All settings except for the initial learning rate (which we set to 0.12) were identical to those from He et al. (41), and we similarly used the detectron2 library for this evaluation. MMCR with 8 views and 100 epochs of pretraining produced an AP50 of 81.9 (this is the mean over three independent fine tunes, the standard deviation was 0.2), demonstrating that the representeations generated by our method are not limited to object recognition; Table **??** gives full results. For reference, a representation trained using MoCo v2 for 200 epochs achieves an AP50 (the most common evaluation metric for this dataset) of 82.4 (in the table below we report the result we obtained for MoCo pretrained for 100 epochs for consistency).

## N  Limitations

Time and compute limitations prevented us from conducting exhaustive optimization of all design choices. For example we do not explore effects of varying the projector network width and depth. We additionally restrict the model to a ResNet-50 encoding network pretrained on the ImageNet-1k

| Method | mAP | AP50 | AP75 |
|--------|-----|------|------|
| Barlow | 53.1 | 80.9 | 57.7 |
| SwAV | 54.4 | 81.6 | 61.0 |
| MMCR | 54.6 | 81.9 | 60.6 |
| SimCLR | 54.7 | 81.7 | 60.2 |
| MoCo v2 | 55.6 | **82.3** | 61.7 |
| BYOL | **56.0** | **82.3** | **62.0** |

Table 6: We follow (41; 77), fine tuning the representation network for detection with a Faster R-CNN head and C-4 backbone on the VOC07+12 dataset.

dataset for 100 epochs. This choice allows us to make fair comparisons to several recently developed SSL methods with a modest compute budget, but precludes answering questions about how the method scales up to larger problems.

We also note that although our objective function has favorable computational complexity compared to existing methods, the evaluation of our objective is not straightforward to distribute across machines. This is because we need to compute the SVD over the centroid matrix, which requires gathering the outputs of a distributed forward pass onto a single machine. Efficient methods for the distributed computation of the SVD could help alleviate this issue in the future.

## O    Impact of Momentum Encoder

With all other settings fixed, we find that the use of a momentum encoder confers a small advantage in terms of classification performance. Below we report frozen linear evaluation accuracy for MMCR training 2, 4, and 8 views with and without a momentum encoder.

|  | 2 Views | 4 Views | 8 Views |
|--|---------|---------|---------|
| With Momentum Encoder | 69.5 | 71.4 | 72.1 |
| Without Momentum Encoder | 68.4 | 70.2 | 71.5 |

Table 7: Shown are Top-1 accuracies under the linear evaluation protocol for representations trained with or without a momentum encoder using the settings described above. It appears that as the number of views increases the added diversity of positive sample representations conferred by the momentum encoder diminishes.

## P    MMCR Benefits from Longer Pretraining

To verify that MMCR benefits from longer pretraining we modified our setup to pretrain with two views and reduced our base learning rate to 0.4 and train for 1000 epochs. We additionally changed the hyperparameters for pretraining the linear classifier when evaluating this model (the classifier was trained with an learning rate of 0.3 and batch size of 2048 for 100 epochs). We did not perform any hyperparameter tuning for this setting, so these results represent a performance floor for 2-view MMCR.

| Method | Top-1 Accuracy |
|---|---|
| SimCLR | 69.3 |
| MoCov2 | 71.1 |
| SimSiam | 71.3 |
| SwAV (without multicrop) | 71.3 |
| MMCR (2 views) | 73.0 |
| Barlow Twins | 73.2 |
| BYOL | 74.3 |
| SwAV (with multicrop) | 75.3 |
| NNCLR | 75.6 |
| ReLICv2 (69) | **77.1** |

Table 8: Shown are Top-1 accuracies under the linear evaluation protocol for representations trained for 1000 epochs using various self supervised learning frameworks.

