# OpenReview forum: "Learning Efficient Coding of Natural Images with Maximum Manifold Capacity Representations"
_NeurIPS.cc/2023/Conference — NeurIPS 2023 poster_

### Official Review · Reviewer_9aX7 · 2023-07-05

**Soundness:** 3 good
**Presentation:** 4 excellent
**Contribution:** 3 good
**Rating:** 6
**Confidence:** 1

**Summary:**

This paper develops a form that facilitates direct optimization, use it to learn Maximum Manifold Capacity Representations (MMCRs), and demonstrate that these are competitive with state-of-the-art results on standard self-supervised learning (SSL) recognition benchmarks. Empirical analyses reveal important differences between MMCRs and the representations learned by other SSL frameworks.

**Strengths:**

Inspired by manifold capacity theory, the proposed self-supervised learning algorithm is efficient to learn and requires neither large batch size nor large embedding dimension.

This work also provides very interesting empirical results which are well aligned with the proposed theorem.

Besides, the paper is well-presented.

**Weaknesses:**

Since I am not in this area, the things mentioned in this paper are very convincing to me. I cannot point out very specific weakness.

**Questions:**

NA

---

> ### Author Rebuttal · Authors · 2023-08-10
>
> Thank you for your positive evaluation of our work. Please let us know if any questions arise during the discussion period and we will be happy to clarify.

---

### Official Review · Reviewer_g5Gb · 2023-07-06

**Soundness:** 2 fair
**Presentation:** 2 fair
**Contribution:** 2 fair
**Rating:** 6
**Confidence:** 4

**Summary:**

This paper proposes a new SSL objective to directly optimize manifold capacity (num of categories represented in a linearly separable fashion). It is found that the resulting representations can support linear classification well. It is also shown that the representations are somewhat sample efficient (reasonable performance in 1% and 10% settings, Food, Flowers, etc). It is also found that that the representations are slightly better predictors of neuronal representations in V2 compared to baselines.

**Strengths:**

1. The description of the objective function and simplifying assumptions are clear
2. The evaluations (e.g. Tables 1, 2) and analyses (e.g. Figs, 4, 5) are clear.
3. The eigenspectrum alpha being close to 1 is an interesting finding.





**Weaknesses:**

[Limited evaluations, objective too specific?]

There is some novelty in the specific objective function used, but evaluation is limited. For example, by more directly optimizing for linear separability has there been a loss in generality of representations, e.g. on segmentation or depth estimation? The detection results in the appendix appear to be consistent with MMCR being more specialized for linear classification. Also unclear how well does the objective work when the data are more complex (e.g. COCO)? Imagenet normally contains few objects and the correct category is very often an object in the center. Moreover, the architecture explored is also limited to ResNet-50 and is not competitive with more recent SSL methods (e.g. RELICv2, NNCLR).

Another important limitation is that evaluations are for SSL models only trained for 100 epochs. Different SSL methods converge to good representations at different speeds, so limiting evaluation to 100 epochs is does not give us an adequate understanding of various methods - this limitation applies to comparisons on ml benchmarks as well as neuronal data.

[Purpose of comparing to neural data unclear.]

MMCR has slightly better predictive power in V2, and the eigenspectrum alpha is close to 1. These are interesting empirical observations, but the discussion does not appear to really engage with these. In the discussion (l316) it is claimed that future work should go beyond accounting for current data. It is not clear to me how the current work provides an account of current data? What are the precise claims/hypotheses being made and how do the experiments support it? What does it entail for our understanding of the brain that was not previously known?


**Questions:**

1. Can the authors please make their discussion on relevance to neuroscience more specific?
2. Can the authors please expand on their evaluations along any of the directions listed in weaknesses?

---

> ### Author Rebuttal · Authors · 2023-08-10
>
> Thank you for your review of our work, our responses to the listed weaknesses are below:
>
> - *[Limited evaluations, objective too specific?]*: We agree that evaluating on more tasks can strengthen the assessment of a learned representation. We have run additional evaluations of baseline models trained in the same setting (100 epochs), which show that MMCR performs comparably to other SOTA methods on object detection and outperforms the supervised baseline. These evaluations are on the VOC dataset using the same procedure described in the appendix of our original submission:
>
> |             |  mAP  |  AP50  |  AP75  |
> |-------------|-------|--------|--------|
> | Supervised  |  53.5 |  81.3  |  58.8  |
> | MMCR        |  54.6 |  81.9  |  60.6  |
> | SimCLR      |  54.7 |  81.3  |  60.2  |
> | SwAV+multicrop | 54.4 |  81.6  |  61.0  |
> | BYOL        |  56.0 |  82.3  |  62.0  |
> | Barlow Twins |  53.1 |  80.9  |  57.7  |
> | MoCo v2* (200 epochs) | 57.0 |  82.4  |  64.0  |
>
> Results for MoCo are reported from the official paper repository.
>
> - *"Another important limitation is that evaluations are for SSL models only trained for 100 epochs..."*: We chose this setting because implementations and models are readily available using ResNet-50 with 100 epochs of pretraining, allowing us to compare to many existing methods directly while operating with a limited compute budget. We do agree that it would be valuable to demonstrate that our method’s strong performance is not limited to 100 epoch pretraining, and have now launched experiments using 1000 epochs and 2 views. This experiment is still running at the time of posting but we will include results in the updated paper.
>
>
>     It is worth noting that NNCLR does report results for linear evaluation using 100 epochs of pretraining, where they achieve 69.4%    accuracy, slightly underperforming MMCR with two views.
>
>
> - *[Purpose of comparing to neural data unclear.]*: The results show that MMCR is competitive in explaining physiological data from area V2, but the BrainScore evaluation does not provide a means of assessing/interpreting the detailed nature of the fits. Our table indicates that, although different objective functions produce representations with very different spectral properties (as evidenced by the spread in participation ratio and alpha decay coefficients), there is much less spread in their neural predictivity.

---

> > ### Comment · Reviewer_g5Gb · 2023-08-13
> > **Still Confused. Results do not seem favorable to MMCR.**
> >
> > Dear Authors, Thanks for your response.
> >
> > -  "The results show that MMCR is competitive in _explaining_ physiological data from area V2". I'm confused by this. How does MMCR _explain_ the physiological data in V2? Sure, it has predictive power. But it's not clear to me how this _explains_ anything. Overall, I'm still confused by the comparison to neural data and find myself going back to the original question I had, what do we know now about the brain that we din't before?
> >
> > - "..although different objective functions produce representations with very different spectral properties (as evidenced by the spread in participation ratio and alpha decay coefficients), there is much less spread in their neural predictivity." This makes me even more confused, because you are suggesting that the specific (MMCR or SimCLR) SSL objective does not matter very much.
> >
> > -  VOC results. Thanks for reporting these numbers. These seem to confirm that any benefits of MMCR over baseline similar SSL objectives are restricted to the linear classification setting.
> >
> > - Results from longer training and different architectures could potentially strengthen the paper. But even if these numbers are not available, please include stronger baselines in the paper (e.g. NNCLR).

---

> > > ### Comment · Reviewer_moiT · 2023-08-13
> > > **Hopefully clarifying comment about "explaining" physiological data**
> > >
> > > > "The results show that MMCR is competitive in explaining physiological data from area V2". I'm confused by this. How does MMCR explain the physiological data in V2? Sure, it has predictive power. But it's not clear to me how this explains anything. Overall, I'm still confused by the comparison to neural data and find myself going back to the original question I had, what do we know now about the brain that we din't before?
> > >
> > > To chime in, in this context, I believe "explaining physiological data" refers to how much variance the regressions from artificial embeddings to neural recordings explain, in the sense of https://en.wikipedia.org/wiki/Explained_variation and https://en.wikipedia.org/wiki/Fraction_of_variance_unexplained.
> > >
> > > I may be mistaken, but I believe in papers that use these sorts of neural regressions, the terminology "explain" is commonly used, and while I personally understand Reviewer g5Gb's comment that "explain" might be incorrectly suggestive, I don't think we can fault these authors for trying to be consistent with established terminology.
> > >
> > > As a suggested improvement, perhaps the authors could add "variance", e.g., "explaining _variance_ in physiological data from area V2"?

---

> > > > ### Comment · Reviewer_g5Gb · 2023-08-13
> > > > **Clarification about terminology would be helpful indeed, but my confusion with the results is deeper**
> > > >
> > > > Thanks for chiming in moiT. I agree that expanding slightly on what is meant by "explained" in this context can be helpful - especially as many papers in NeurIPS will be using "explain" in quite a different way than here.
> > > >
> > > > But terminology aside, my confusion is deeper. To expand on it - the analyses on neural data don't appear to strengthen the paper. If anything, they appear to weaken the argument for MMCR - instead they suggest (as also noted by the authors above) that it doesn't particularly matter which objective (MMCR or SimCLR) is used, all of them are quite similarly predictive of the neural data. Indeed, recent work [1] suggests that this is to be expected  - the training data is identical, the objectives are trying to do the same thing with minor differences, and the architecture and duration of training is identical. In the end, what have we learned from all of this? What is the hypothesis under consideration - that brain implements this specific objective function, MMCR to optimize representations for linear classification?
> > > >
> > > > It would be very helpful if the authors could expand the discussion of these analyses to be more specific and concrete about what this means for our understanding about the brain.
> > > >
> > > >
> > > > 1.  What can 1.8 billion regressions tell us about the pressures shaping high-level visual representation in brains and machines? Conwell, et al, bioRxiv, 2022.

---

> > > > > ### Comment · Reviewer_moiT · 2023-08-13
> > > > > **Response to Reviewer g6Gb**
> > > > >
> > > > > > I agree that expanding slightly on what is meant by "explained" in this context can be helpful - especially as many papers in NeurIPS will be using "explain" in quite a different way than here.
> > > > >
> > > > > Totally agree. I hope the authors see this and confirm.
> > > > >
> > > > > > the analyses on neural data don't appear to strengthen the paper. If anything, they appear to weaken the argument for MMCR - instead they suggest (as also noted by the authors above) that it doesn't particularly matter which objective (MMCR or SimCLR) is used, all of them are quite similarly predictive of the neural data. Indeed, recent work [1] suggests that this is to be expected - the training data is identical, the objectives are trying to do the same thing with minor differences, and the architecture and duration of training is identical. In the end, what have we learned from all of this? What is the hypothesis under consideration - that brain implements this specific objective function, MMCR to optimize representations for linear classification?
> > > > >
> > > > > I think this is a great question to be asking. I asked myself this and the conclusion I reached for myself is the following:
> > > > >
> > > > > [1] claims that for high performing SSL algorithms in vision, they all do approximately equally well; my interpretation of this paper then is that this geometric manifold separability approach should be added to this class of high performing SSL algorithms. To me, the contribution of this paper is less about creating the best possible SSL algorithm for vision (especially if most algorithms are roughly equivalent) & more about showing that a different lineage of methods, designed for analyzing representations and based on linearly separable manifold capacity calculations, can be converted to a high performing SSL algorithm. I view the rest of the paper as providing intuition for what the algorithm does, showing that performance is comparable to others, but some other characteristics may be different.
> > > > >
> > > > > As I wrote in another comment, my philosophy of what it means to be a reviewer changed at ICML 2023. I feel like reviewers are pretty noisy as a population; we're not very good at identifying impactful papers. Consequently, I feel like our standard should be: is this paper sufficiently well put together that others in our field could take a look without considering their time wasted? If so, put the paper out there and let the community decide its significance. Based on this standard, I feel like this paper should be accepted. You may of course have a different (and probably higher) standard.
> > > > >
> > > > > I also urge the authors to chime in as well.

---

> > > > > > ### Comment · Reviewer_g5Gb · 2023-08-13
> > > > > >
> > > > > > Thanks for sharing your perspective on this. My (current) perspective is that this work in it's current form does not meet even this lower bar. The field is inundated with methods from different lineages that do very similar things and if the summary of experiments from a work is that it is competitive with some baselines with small differences in lineage being the only distinguishing factor, then it might be worth iterating further on the work before being shared with the community.

---

> > > > > ### Author Response · Authors · 2023-08-14
> > > > > **Clarifications**
> > > > >
> > > > > As suggested by reviewer moIT, we can confirm that by using “explain” we indeed meant “variance explained.” We will clarify this terminology in the revision.
> > > > >
> > > > > Regarding the insights gleaned from our incorporation of neural data: In short we present findings that are in direct support of the larger scale study [1]. Specifically we contribute additional evidence to the claim from [1]: “that there is meaningful diversity in the learned representations of models that our metrics are failing to translate into significantly different brain-predictivity scores.”
> > > > >
> > > > > This study did not appear on the archive in its current form until after the submission of this work (with important additions including the analyses on effective dimensionality, and the conclusion that data-diet is an extremely important factor for neural predictivity). Furthermore, that study focuses on the prediction of voxel responses in human fMRI encoding while our comparison is to electrophysiological measurements in macaque, so our study provides new and complementary evidence from a different domain in support of the same conclusion. An additional point of novelty in our work is the inclusion of the alpha decay coefficient in such analyses. We propose to insert the following paragraph at the conclusion of section 3.3 to clarify the contribution of our experiments and better connect to other similar sentiments that appear in the literature:
> > > > >
> > > > > “These results highlight the promise of geometrical properties as metrics to better distinguish between models of cortical computations than existing methods do. We find that in this controlled setting each model explains a very similar fraction of neural variance through linear regression. This is consistent with recent and concurrent works, [1, 2] which have identified this lack of ability to discriminate between alternative models as a weakness of the dominant paradigm used for model-to-brain comparisons. However, our results demonstrate that different SSL algorithms produce representations with meaningfully different geometries (as evidenced by the large spread in the spectral properties such as the participation ratio and decay coefficient). This suggests the need for the development of new metrics for comparing models to data, such as geometrical measures, that capture these important differences between candidate models.”
> > > > >
> > > > > [1] What can 1.8 billion regressions tell us about the pressures shaping high-level visual representation in brains and machines? Conwell, et al, bioRxiv, 2023.
> > > > >
> > > > > [2] If deep learning is the answer, what is the question? Saxe, et al., Nature Reviews Neuroscience, 2021

---

> > > > > > ### Comment · Reviewer_g5Gb · 2023-08-14
> > > > > >
> > > > > > Dear authors, thanks for the clarifications. The paragraph you are proposing to add sounds good to me and can be a good discussion to have at the conference with the community. Please also add other numbers (VOC) as well as stronger baselines (e.g. NNCLR). I've updated my score to reflect these changes.

---

> > > ### Author Response · Authors · 2023-08-14
> > >
> > > We agree that the evaluations suggest MMCR is strongest in the linear classification setting, though our method is also competitive with baselines in terms of detection. We will also include NNCLR as an additional baseline as per your suggestion.
> > >
> > > We feel that the introduction of a new objective function that is competitive with several baselines across a variety of tasks and produces a representation with measurably different properties warrants inclusion in the literature.

---

### Official Review · Reviewer_eLMY · 2023-07-06

**Soundness:** 2 fair
**Presentation:** 2 fair
**Contribution:** 3 good
**Rating:** 5
**Confidence:** 4

**Summary:**

This paper proposes an alternative way (i.e., maximizing manifold capacity) to learn useful representation in a self-supervised manner. In particular, it maximizes the separability of the centroid manifold (i.e., negative samples), while the second term aims to minimize the nuclear norm among positive samples.
tends to figure out crucial properties of self-supervised learning (SSL) methods, which promote good performance on downstream tasks. To reach this target, the authors propose a unifying conceptual ISSL framework. In particular, their main contributions contains: i) increasing the dimensionality of presentation and using asymmetric projection heads; ii) presenting a new non-contrastive ISSL objective; iii) applying non-linear probes.

**Strengths:**

This paper introduces manifold capacity in self-supervised learning (SSL). This idea is somewhat novel and provides an alternative way to avoid the collapse issue in SSL. Extensive experiments conducted on nearly 7 datasets demonstrate that the proposed method achieves comparable performance to existing state-of-the-art (SOTA) methods. Furthermore, empirical analysis of Maximum Manifold Capacity Representations (MMCRs) reveals distinct characteristics compared to existing approaches.

**Weaknesses:**

However, my main doubts/concerns regarding the paper are the following:

- As shown in Line 138-141, the first term aims to maximize the separability of negative samples, while the second term aims to minimize the nuclear norm among positive samples. The motivation behind the proposed loss is similar to [1], which pursues "the alignment of features from positive pairs and the uniformity of the induced distribution of the normalized features on the hypersphere." It would be better to provide further discussion in relation or difference to [1].
- The observation in Figure 6 is interesting. However, this result raises the question of whether implicitly minimizing object manifold compression is truly necessary, as the second term with nuclear norm does not impact the mean manifold nuclear norm. Therefore, conducting an ablation study between these two terms on several downstream tasks is essential to further investigate their effects.
- The proposed MMCR method shares strong connections with nuclear norm-based approaches [2,3,4], especially in relation to [2]. However, the current study lacks either theoretical or empirical comparisons between MMCR and these nuclear norm-based methods.
- In Figure 2, the Mean Filed Manifold Capacity Analysis shows that existing SOTA methods with different objectives all explicitly learn representations with large radii but low dimensionality. These results may question the necessity of the proposed method.
- In Figure 5, the centroid similarity distributions of MMCR for both same class and distinct classes appear to be generally smaller than those of existing methods (Barlow Twins and SimCLR). Therefore, this comparison may be somewhat unfair. To ensure a fair comparison, it would be better to normalize the centroid similarities (e.g., by computing z-scores) for each method and then compare the normalized distribution across different methods.

Minor:
- The experimental setting described in Section 3.2 is unclear, which could potentially result in unfair comparisons between MMCR and other state-of-the-art (SOTA) methods. If MMCR employs multi-crop augmentation while Barlow Twins and SimCLR only use two views, the comparison would be unfair in Figure 2 and 5.
- The authors might unintentionally modify the default LaTeX template, which results in incorrect formatting of the citations. In particular, (1) -> [1] or (1;2) -> [1,2].
- Incorrect format of theorems, lemmas, and proofs is detrimental to this paper since it makes the paper appears informal.
- Almost all equations lack commas or full stops at the end.
- Please standardize the usage of either "Fig. x" or "Figure x" for consistency.
- Figure 2: "see E" -> "see appendix E". Table 2: "see L" -> "see appendix L".


[1] T. Wang, et al., "Understanding Contrastive Representation Learning through Alignment and Uniformity on the Hypersphere", ICML 2020.

[2] Y. Wang, et al., "A Low Rank Promoting Prior for Unsupervised Contrastive Learning", TPAMI 2022.

[3] O. Hénaff, et al., "The Local Low-dimensionality of Natural Images", ICLR 2015.

[4] J. Lezama, et al., "OLÉ: Orthogonal Low-rank Embedding, A Plug and Play Geometric Loss for Deep Learning", CVPR 2018.

**Questions:**

The questions are corresponding to the above main concerns:
- What is the relation or difference between the proposed objective and [1]?
- Please add more theoretical or empirical analyses on the necessity of implicitly minimizing object manifold compression.
- Please add more theoretical or empirical comparisons between MMCR and the nuclear norm-based methods [2,3,4].
- Please add more discussion about the results in Figure 2.

**Limitations:**

Yes, the authors have addressed the societal impact of their work.

---

> ### Author Rebuttal · Authors · 2023-08-10
>
> Thank you for your thorough review of our submission. Below we respond to each of the listed weaknesses:
>
> - *"As shown in Line 138-141..."*: Many self-supervised learning methods share these core motivations, and [1] in particular demonstrates that the logarithm of the average pairwise Gaussian potential is optimized by a uniform distribution on the hypersphere. The unique aspect of our method is that it doesn’t rely on pairwise comparisons in order to achieve the desired properties in the global representation, but rather optimizes a population level metric directly. We will make this point more explicit in the revision.
>
> - *"The observation in Figure 6 is interesting..."*: We feel that Figure 6 does not imply the necessity of implicit manifold compression, but rather demonstrates the effectiveness of doing so. This is an important interpretational difference, as using explicit compression substantially increases the computational complexity of evaluating the objective. In practice we observe that including this term in the loss significantly increases the run time, and on CIFAR experiments we did not observe any benefit in terms of the quality of the learned representation. We will emphasize this practical advantage of implicit compression in the revised paper.
>
> - *"The proposed MMCR method shares strong connections"*: Our formulation is related to [2, 3, 4] in the use of nuclear norm (and we’ve cited all of them for this). However, each of them differ significantly enough from our approach that a more detailed comparison did not seem warranted.  Specifically:
>     - [3] is primarily focused on learning a set of linear filters in an autoencoder architecture. The substantial difference in experimental setting prevents a useful empirical comparison, but we will include a more thorough discussion of this paper in our revised related works section.
>     - [2, 4] employ low rank (nuclear norm) regularizers to supplement their objective functions. In contrast, our objective directly maximizes rank. In addition, [4] differs in that it employs supervised training. We do think a more direct comparison with [2] could be interesting, i.e. we could compare the spectra of augmentation manifolds and the global dimensionality of the learned representations. A fair comparison will require matching the pre-training settings (specifically the amount of pretraining, LORAC trains for 200 epochs), and compute limitations have prevented us from completing these experiments during the rebuttal period. We will however report preliminary results during the upcoming discussion period and in include such analyses in the final paper.
>
> - *"In Figure 2, the Mean Filed Manifold Capacity Analysis shows..."*: For methods that perform strongly in terms of linear classification, capacity analysis provides further insight as to whether this is accomplished with class manifolds with low dimensionality or low radius. Figure 2 demonstrates that MMCR yields high capacity by reducing dimensionality at the expense of increasing radius, relative to SOTA methods.
>
>     Your comment helped us to realize that the formatting of Figure 2 obscures this point, since the differences only emerge at the tail end of the ResNet hierarchy. In the revision, we’ll include a table of values taken from the final representation layer, and move the figure to the appendix.
>
> - *"In Figure 5, the centroid similarity distributions..."*: We would appreciate it if the reviewer could clarify this concern. The distribution of similarities has a smaller mean than the alternatives, which is the message we were trying to communicate. If the distributions for each method were z-scored, they would differ only in higher order (shape related) properties, which could be interesting but is not central to the argument being made.
>
> - *"The experimental setting described in Section 3.2..."*: All the model evaluations use the same number of augmentations - we will clarify this point in the revised paper. It is true that during *pre-training*, SimCLR and Barlow Twins use 2-view augmentations.  We think this is fair, since these were the procedures used in their respective original implementations (in addition, it is not clear how Barlow Twins would be adapted to the multi-view setting).  We’ll clarify these points in the revised text.
>
> - *[Minor]*: Thank you for bringing these formatting issues to our attention, these will be corrected in the revised text.

---

> > ### Comment · Reviewer_eLMY · 2023-08-15
> > **Official Comment by Reviewer eLMY**
> >
> > Thanks for your response. The comments from the authors solve most of my concerns, and I have updated the score.
> >
> > Clarification to "In Figure 5, the centroid similarity distributions...": In Figure 5, the experimental settings for the three methods differ significantly, which will result in different distributions of centroid similarity. Therefore, making a direct comparison of these distributions might not provide an apples-to-apples comparison, which will weaken the claim.

---

### Official Review · Reviewer_1Hrd · 2023-07-06

**Soundness:** 3 good
**Presentation:** 3 good
**Contribution:** 3 good
**Rating:** 7
**Confidence:** 3

**Summary:**

The efficient coding hypothesis suggests sensory systems maximize mutual information between their inputs and the environment. A recent adaptation, "manifold capacity", calculates the number of linearly separable object categories, but is computationally intensive. The authors simplify this measure to directly optimize Maximum Manifold Capacity Representations (MMCRs). They show MMCRs are competitive with top results on self-supervised learning (SSL) recognition benchmarks. MMCRs differ from other SSL frameworks and may enhance class separability through manifold compression. On neural predictivity benchmarks, MMCRs prove competitive as models of the ventral stream, a part of the brain involved in object recognition.

**Strengths:**

-	The paper presents a novel method on SSL leveraging Manifold Capacity theory. The contribution is clear and precise.
-	The paper connect SSL with Manifold Capacity theory with a effective theoretical formulation for the SSL objective.
-	The paper empirically validates its claim by running large scale experiments and demonstrates competitive performance.
-	The method mitigate the large batch / large dimension training bottleneck of the previous SSL method.
-	The paper also evaluates on the neural data to show that its method can explain the neural data better.


**Weaknesses:**

-	Although the initial results look very promising, the author didn’t include an error bar in Table-1. It would be beneficial to see the error bar to further eliminate the effect of chance.
-	The paper posits that sensory systems maximize mutual information between their representations and the environment, based on the efficient coding hypothesis. However, it might lack a discussion on the biological plausibility of MMCRs. How feasible is it for biological sensory systems to implement MMCRs? How does this model account for the biological constraints mentioned in the efficient coding hypothesis?
-	[Minor] In the legend of Figure 2, the description indicates the top row showing the radius but the actual graph shows dimensionality.


**Questions:**

- An illustrative figure that intuitively demonstrate the intuition of singularity of the covariance matrix in terms of the objective function would be beneficial to the reader's understanding.
- In the introduction, it might be better to highlight the contribution to the SSL more clearly.

**Limitations:**

yes

---

> ### Author Rebuttal · Authors · 2023-08-10
>
> Thank you for your positive review of our paper. We respond to each of the listed weaknesses below:
>
> - *"Although the initial results look very promising..."*: We agree including a measure of uncertainty would help strengthen the paper. For ImageNet experiments repeated trainings of the linear classifier yields a standard deviation of \~0.05, and for the smaller datasets we have observed slightly more variability (~0.1). We will update the table with these confidence intervals in the revised paper.
>
> - *"The paper posits that sensory systems..."*: Thank you for this thoughtful question. One of the key biological constraints is that not all of the information carried by a sensory stream can be faithfully encoded with limited resources. This constraint is applied in MMCR by encouraging invariance to augmentation variability. The learned representations are efficient in that they compress this source of variability while maximizing the discriminability over the images in the dataset.
>
>     Moreover, we speculate that the MMCR objective is better suited to biological implementation because it does not require a large number of pairwise comparisons to evaluate the objective. Instead the MMCR objective is a function of the summary statistics (singular values) of a global representation of the image manifold. Whether and how this could be optimized using neural circuits is of interest, but well beyond the scope of the current paper.
>
> - *"In the legend of Figure 2..."*: Thank you for bringing this to our attention, we will revise the caption in the updated paper.

---

### Official Review · Reviewer_moiT · 2023-07-07

**Soundness:** 3 good
**Presentation:** 3 good
**Contribution:** 3 good
**Rating:** 7
**Confidence:** 4

**Summary:**

The authors convert recent results in manifold perceptron capacity calculations to a practical self-supervised learning algorithm. They then (1) provide one theoretical result, (2) show their method is predictive of primate visual cortex recordings, and (3) characterize some empirical properties of their method versus other methods.

**Strengths:**

- The conversion of manifold perceptron capacity to a practically implementable SSL algorithm is novel

- This paper is quite thorough. It contains a little bit of everything: a new algorithm, one theoretical result for the algorithm, comparison with biological data, empirical study of how the algorithm differs from other SSL algorithms, eigenspectrum decay. Its experimental results are particularly thorough.

**Weaknesses:**

Ordered by location in the text, rather than prioritization:

- Section 2.1 either (1) makes a subtle but important leap that I'm not sure I followed or (2) uses confusing pluralization. Lines 93 to 103 state we consider P manifolds (plural) and then discuss three quantities: (1) the manifold radius (singular), (2) the manifold dimension (singular), (3) the correlation between the manifold centroids (plural). Additionally, a subscript $M$ appears and I do not know what this $M$ refers to. I'm not trying to nitpick pluralization; rather, I'm confused whether each of the P manifolds has its own radius and its own dimension, or whether all P manifolds collectively have 1 radius and 1 dimension. If the former, how are each of the P manifolds' radii and dimensions combined to determine the threshold capacity of the manifold population? If the latter, what does linear separability mean for a monolithic grouping of the P manifolds? Or must all $P$ manifolds have the same radius and dimension, but are permitted to possess their own centroids? I would appreciate if the authors could clarify this section.

- The move from Section 2.1 to Section 2.2 needs better motivation. Section 2.1 introduces the manifold capacity as $\phi(R_M \sqrt{D_M})$ with differentiable, closed-form expressions for both $R_M$ and $D_M$. I am then expecting that we'll be maximizing capacity by minimizing $R_M \sqrt{D_M}$. But instead, we jump to Equation (2)'s, with a small inline connection to $\phi(R_M \sqrt{D_M})$, and are then told that the second term of Equation 2 doesn't matter and so we'll set its prefactor $\lambda$ to 0. I think this section could be greatly improved if the authors emphasize & explain the inline approximation , and better motivate Equation 2. I would recommend either leaving out the second term of Eqn 2 altogether, or alternatively, clarifying why the second term _should_ be mentioned if it is to be immediately eliminated.

- I'm unsure whether I should be persuaded by the 2D example of 2 centroids (Equations 3 and 4) . I do love attempting to build intuition, and I understand that no closed form expressions exists for singular values of an arbitrary matrix, so I commend the authors, but I'm unsure whether the intuition will hold in (a) higher dimensions and (2) with significantly more data i.e. more centroids. Even staying in 2D, what will MMR do if we have 3, 4, 5 etc. centroids? I might've guess something like the Thomson problem (https://en.wikipedia.org/wiki/Thomson_problem) but that can't be the case because 2 electrons choose antipodal points where MMCR chooses orthogonal points.

- Moreover, upon another read, I realized the paragraph "Compression by Maximizing Centroid Nuclear Norm Alone" seems odd compared with Equation 2. The paragraph argues that the nuclear norm of $C$ alone is sufficient because it both (1) incentivizes making each manifold as tightly compressed as possible (i.e. each mean should have maximum norm, bounded above by 1) and (2) incentivizes making centroids pairwise orthogonal. I agree with this - the 2D case is clear and convincing. But if those two incentives are what we care about, why not directly use them as our loss? I'm specifically suggesting something like $-\sum_b ||c_b||^2 + \sum_{b, b'} (c_b \cdot c_{b'})^2$? And if so, isn't that almost exactly the loss of TiCo (Zhu et al 2021/2022 https://arxiv.org/abs/2206.10698 ), specifically equation 6?

- With all SSL papers, there are many implementation decisions and hyperparameters that complicate interpreting the results. Line 205-208 caught my eye ("We additionally employ a momentum encoder for ImageNet pre-training...". While there is nothing wrong with using a momentum encoder, I would like to know what performance MMCR achieves with vs. without the momentum encoder. Could the authors please include this ablation? If I missed it, I apologize.

- Please add some measure of uncertainty (e.g. 95% confidence intervals) to Figure 3 (unless such intervals already exist and are too small to be seen, in which case, please note that in the figure caption).

- The authors write "Elmoznino and Bonner (23) found that high dimensionality in ANN representations was associated with ability to both predict neural data." and later "Additionally, Elmoznino and Bonner (23) report that high intrinsic dimensionality (as measured with the participation ratio of the representation covariance) is correlated with a representation’s ability to predict neural activity." The claim that participation ratio is highly correlated with neural predictivity was noted concurrently by Schaeffer et al. 2022 "No Free Lunch from Deep Learning in Neuroscience" at an ICML 2022 workshop (albeit in mouse MEC on navigation tasks, rather than primate visual cortex on vision tasks) and published at NeurIPS 2022. Then, soon after, Tuckute et al. 2022 "Many but not all deep neural network audio models capture brain responses and exhibit hierarchical region correspondence" released a preprint showing the same correlation between participation ratio and neural predictivity in human fMRI recordings of audio (Figure S6 in biorxiv v1; I'm too lazy to find the same plot in the more recent biorxiv v4). From an attribution perspective, I believe all three should be cited since they were all within ~1-3 months of one another. I also think citing all three will strengthen your paper since it points out the same phenomenon appears in multiple modalities (navigation, vision, audition), multiple species (rodent - can't remember mouse or rate, monkey, human) , multiple recording technologies.

I realize that many of these "weaknesses" might be my own misunderstandings. If the authors could clarify, or lightly revise the paper to address these above points in a satisfactory manner, I would be happy to increase my score :)

**Questions:**

- Figure 5: The authors state "the InfoNCE loss employed in Chen et al. (12) benefits when negative pairs are as dissimilar as possible, which is achieved when the two points lie in opposite regions of the same subspace". If this is correct (and I believe it is), we should expect to see negative cosine similarities between distinct classes. But then the right subplot shows that SimCLR has positive cosine similarity (~0.3) between distinct classes. Could the authors please clarify?

---

> ### Author Rebuttal · Authors · 2023-08-10
>
> Thank you for your very thorough review of our work! Below we address each point listed in the weaknesses:
>
> - *"Section 2.1 ..."*:   Apologies that this was not expressed more clearly: each of the manifolds is assumed to have the same dimensionality and radius, but each has its own centroid. We will clarify in the revision.
>
> - *"The move from section 2.1 to section 2.2..."*: The jump to Equation (2) is a bit abrupt (we left these steps out due to space constraints). The capacity is a monotonic function of the nuclear norm in lines 114-115.  We neglected to mention that its functional form is not well suited to gradient descent (evaluation requires numerical integration) but the monotonicity means it is largely irrelevant for the purpose of gradient-based optimization. We included the second term in Eq (2) because we were aiming to formulate a contrastive objective: an invariance term (the second) and a term to prevent representational collapse (the first). It was not initially obvious to us that the collapse prevention term (in tandem with the unit sphere constraint) naturally encourages invariance. We will revise the presentation to communicate the above sentiment “up-front,” and then modify Eq (2) to only include the first term.
>
> - *"I'm unsure whether I should be persuaded by the 2D example... "*: Thank you for carefully considering the implications of our proposed objective. We included the 2-D example to build intuition for two properties of the objective: it maximizes the angle between the centroids, and it also encourages augmentation manifold compression (indirectly, by maximizing the norm of centroids). When there are more centroids than dimensions, maximizing the nuclear norm encourages them to form a simplex equiangular tight frame (sETF) - note that this can only be achieved exactly for certain combinations of dimensionality and number of points.  For the 2D case, they would be equally distributed around  the unit circle.
>
> - "Moreover, upon another read..."*: The most important property of our objective is that it incentivizes global high dimensionality without relying on expectations over large numbers of pairwise comparisons (as required by TiCo and many other SSL methods). We will clarify this in the revised text.
>
> - *"With all SSL papers..."*: Thanks for the suggestion - we agree this is worth reporting. The momentum encoder conferred a small advantage in terms of downstream classification performance:
>
> |                             | 2 Views | 4 Views | 8 Views |
> |---------------------|---------|---------|-------------|
> | With Momentum Encoder |   69.5  |   71.4  |   72.1  |
> | Without Momentum Encoder | 68.4 |   70.2  |   71.5  |
>
> - *"Please add some measure of uncertainty..."*: Thank you for pointing out this omission. The figure does indeed have confidence intervals which are too small to be seen (due to the large sample size). We will mention this information to the figure caption.
>
> - *"The authors write..."*: Thank you for the pointers - we agree that these articles should be cited as well!
>
> - *"Figure 5: The authors state"*: We are visualizing the cosine similarity at the output of the encoder network, which is the learned representation used for downstream tasks. Because the encoder is a ResNet-50, the outputs are rectified and the minimum possible similarity is zero. If we instead look at the outputs of the projector network (where the loss is applied), your intuition is correct: the distribution of centroid similarities for SimCLR for distinct classes peaks below zero (see the figure included in the pdf uploaded along with our general response, we match the setting of Figure 5 except for we instead examine the outputs of the projector for SimCLR).

---

> > ### Comment · Reviewer_moiT · 2023-08-10
> > **Reposting comment that I somehow accidentally made invisible to the authors**
> >
> > Hi! I'll read and respond to your rebuttal later, but I realized just now that a comment was not visible to the authors even though it was meant to be. I posted a comment during the review-writing stage, and I thought the authors weren't included under the comment's Readers because we were still in the review-writing stage, but I now see that hasn't changed, and so the authors might not have seen the comment. Consequently, I'm reposting the comment so that the authors can hopefully see:
> >
> > ===============================
> >
> > My comment in my review, asking how MMCR behaves in 2D with more than 2 centroids, prompted me to sit down and write a small simulation to understand the answer myself. Here, I'm assuming that all we need to focus on are centroids. The code is short and can be run locally on a personal machine in seconds.
> >
> > ```import matplotlib.pyplot as plt
> > import numpy as np
> > import torch
> > import torch.nn as nn
> > import torch.optim as optim
> >
> >
> > # Define the nuclear norm as the sum of singular values
> > def nuclear_norm(matrix):
> >     u, s, v = torch.svd(matrix)
> >     return torch.sum(s)
> >
> >
> > def minimize_nuclear_norm(num_centroids: int) -> np.ndarray:
> >     # Generate N random vectors on the unit circle
> >     # Shape: (N, 2)
> >     centroids = torch.normal(mean=0, std=1, size=(num_centroids, 2))
> >     centroids = centroids / torch.norm(centroids, dim=1, keepdim=True)
> >
> >     # Create a PyTorch variable to hold the matrix
> >     matrix = nn.Parameter(centroids)
> >
> >     # Define the optimizer.
> >     optimizer = optim.Adam([matrix], lr=0.005)
> >
> >     # Optimization loop
> >     for i in range(2500):
> >         optimizer.zero_grad()
> >         loss = -nuclear_norm(matrix)
> >         loss.backward()
> >         optimizer.step()
> >         # Put the row vectors back on the unit circle.
> >         matrix.data = matrix.data / torch.norm(matrix.data, dim=1, keepdim=True)
> >         print('Iteration: {}, loss: {}'.format(i, loss.item()))
> >
> >     # Retrieve the optimized matrix
> >     matrix = matrix.detach().numpy()
> >
> >     return matrix
> >
> >
> > for num_centroids in range(2, 8):
> >     print('N: {}'.format(num_centroids))
> >     optimized_matrix = minimize_nuclear_norm(num_centroids=num_centroids)
> >     plt.close()
> >     fig, ax = plt.subplots(figsize=(6, 6))
> >     for i in range(num_centroids):
> >         ax.plot([0, optimized_matrix[i, 0]], [0, optimized_matrix[i, 1]])
> >     ax.set_ylim(-1.1, 1.1)
> >     ax.set_xlim(-1.1, 1.1)
> >     ax.set_aspect('equal')
> >     plt.show()
> >
> > ```
> >
> > My observations:
> > 1. As the authors report, when there are 2 centroids, then indeed, the centroids move to be orthogonal to one another.
> > 2. 3 centroids will occasionally be equally spaced around the unit circle OR will occasionally lie together within 1 quadrant of the unit circle
> > 3. With more centroids e.g., 4 or 5, MMCR oftentimes place two or more centroids nearly or directly on top of one another.
> >
> > I can't figure out how to upload images to OpenReview, so you may need to run the code yourself a few times to see these various outcomes. Obviously, this was just something quick and I have not quantified the behavior of MMCR over many runs, and I may have done something wrong.
> >
> > Have I done something wrong? If not, it appears that once MMCR is has more centroids than dimensions, MMCR fails to distinguish a subset of centroids from other centroids. Could the authors comment or explore this behavior in depth?  This seems like a bad property for a SSL method to possess.
> >
> > Edit: To follow up, suppose you have C centroids in N dimensions, in an optimal configuration i.e. the nuclear norm is maximized. Suppose you then add one more centroid. Where is the optimal place to put the new centroid? If C>N, then adding the new centroid can't affect the rank, so you still have same number of singular values to optimize. I think the answer is atop an existing centroid, because doing so has zero impact on the nuclear norm and the nuclear norm was previously optimal. Does this make sense or am I mistaken? If so, is stacking centroids one-atop-another desirable behavior?

---

> > > ### Author Response · Authors · 2023-08-11
> > >
> > > Thanks for the code and simulations.  Our interpretation is that sETF solutions are optimal, but there are other configurations that achieve the same nuclear norm. For example, in the 2D case of 4 centroids, both the equally spaced solution and a duplicated pair of orthogonal vectors achieve the optimal loss. So the solution in this case comes down to initialization. But in spaces of reasonably high dimensionality, we believe these degenerate solutions become quite unlikely.  To demonstrate, we examined mean and maximum absolute cosine similarity between optimized centroids, as a function of dimensionality and undercompleteness.  We can no longer include the graphs in a PDF, but our code is attached below (it is only a slight modification of your simulation).  Note that, for example, for 512-D (the dimensionality used in the paper) and 8x under-completeness, the centroids are all pairwise nearly orthogonal (max similarity of \~0.2 out of 4096 choose 2 pairs).
> > >
> > > ```
> > > import numpy as np
> > > import matplotlib.pyplot as plt
> > >
> > > import torch
> > > import torch.nn as nn
> > > import torch.optim as optim
> > > import matplotlib.pyplot as plt
> > > from tqdm import tqdm
> > >
> > > # Define the nuclear norm as the sum of singular values
> > > def nuclear_norm(matrix):
> > >     u, s, v = torch.svd(matrix)
> > >     return torch.sum(s)
> > >
> > >
> > > def minimize_nuclear_norm(num_centroids: int, dim: int) -> np.ndarray:
> > >     # Generate N random vectors on the unit circle
> > >     # Shape: (N, D)
> > >     centroids = torch.normal(mean=0, std=1, size=(num_centroids, dim))
> > >     centroids = centroids / torch.norm(centroids, dim=1, keepdim=True)
> > >
> > >     # Create a PyTorch variable to hold the matrix
> > >     matrix = nn.Parameter(centroids)
> > >
> > >     # Define the optimizer.
> > >     optimizer = optim.Adam([matrix], lr=0.005)
> > >
> > >     # Optimization loop
> > >     losses = []
> > >     for i in tqdm(range(2500)):
> > >         optimizer.zero_grad()
> > >         loss = -nuclear_norm(matrix)
> > >         loss.backward()
> > >         optimizer.step()
> > >         # Put the row vectors back on the unit circle.
> > >         matrix.data = matrix.data / torch.norm(matrix.data, dim=1, keepdim=True)
> > >         #print('Iteration: {}, loss: {}'.format(i, loss.item()))
> > >         losses.append(loss.item())
> > >
> > >     # Retrieve the optimized matrix
> > >     matrix = matrix.detach().numpy()
> > >
> > >     return matrix, losses
> > >
> > > def get_pairwise_similarities(matrix):
> > >   # matrix contains unit norm vectors
> > >   sims = matrix @ matrix.T
> > >
> > >   # extract the upper triangular part
> > >   return sims[np.triu_indices_from(sims, k=1)]
> > >
> > > dims = [2, 4, 8, 16, 32, 64, 128, 256, 512]
> > > centroids_per_dim = [1, 2, 4, 8]
> > >
> > > # sweep over dimensionality and number of centroids per dim
> > > mean_abs_sims = []
> > > max_abs_sims = []
> > > for dim in dims:
> > >   for cpd in centroids_per_dim:
> > >     matrix, losses = minimize_nuclear_norm(num_centroids=dim*cpd, dim=dim)
> > >     sims = get_pairwise_similarities(matrix)
> > >     abs_sims = np.abs(sims)
> > >
> > >     mean_abs_sims.append(np.mean(abs_sims))
> > >     max_abs_sims.append(np.max(abs_sims))
> > >
> > > fig, axs = plt.subplots(1, 2, figsize=(12, 5))
> > >
> > > x = centroids_per_dim
> > >
> > > axs[0].plot(x, mean_abs_sims[:4], label='D=2')
> > > axs[0].plot(x, mean_abs_sims[4:8], label='D=4')
> > > axs[0].plot(x, mean_abs_sims[8:12], label='D=8')
> > > axs[0].plot(x, mean_abs_sims[12:16], label='D=16')
> > > axs[0].plot(x, mean_abs_sims[16:20], label='D=32')
> > > axs[0].plot(x, mean_abs_sims[20:24], label='D=64')
> > > axs[0].plot(x, mean_abs_sims[24:28], label='D=128')
> > > axs[0].plot(x, mean_abs_sims[28:32], label='D=256')
> > > axs[0].plot(x, mean_abs_sims[32:36], label='D=512')
> > > axs[0].legend()
> > >
> > > axs[1].plot(x, max_abs_sims[:4], label='D=2')
> > > axs[1].plot(x, max_abs_sims[4:8], label='D=4')
> > > axs[1].plot(x, max_abs_sims[8:12], label='D=8')
> > > axs[1].plot(x, max_abs_sims[12:16], label='D=16')
> > > axs[1].plot(x, max_abs_sims[16:20], label='D=32')
> > > axs[1].plot(x, max_abs_sims[20:24], label='D=64')
> > > axs[1].plot(x, max_abs_sims[24:28], label='D=128')
> > > axs[1].plot(x, max_abs_sims[28:32], label='D=256')
> > > axs[1].plot(x, max_abs_sims[32:36], label='D=512')
> > >
> > > axs[0].set_xlabel('Centroids Per Dimension')
> > > axs[1].set_xlabel('Centroids Per Dimension')
> > > axs[0].set_ylabel('Absolute Cosine Similarity')
> > > axs[1].set_ylabel('Absolute Cosine Similarity')
> > > axs[0].set_title('Mean Absolute Cosine Similarity')
> > > axs[1].set_title('Max Absolute Cosine Similarity')
> > >
> > > ```

---

> > > > ### Comment · Reviewer_moiT · 2023-08-12
> > > > **Exploring toy simulations**
> > > >
> > > > Thanks for looking into this! I have a couple comments but to make sure I understand your response, I have some small clarification questions as well:
> > > >
> > > > > Our interpretation is that sETF solutions are optimal, but there are other configurations that achieve the same nuclear norm. For example, in the 2D case of 4 centroids, both the equally spaced solution and a duplicated pair of orthogonal vectors achieve the optimal loss.
> > > >
> > > > It seems odd to say that sETF solutions are optimal if non-sETF solutions achieve the same nuclear norm, no?
> > > >
> > > > > So the solution in this case comes down to initialization.
> > > >
> > > > Agreed.
> > > >
> > > > > But in spaces of reasonably high dimensionality, we believe these degenerate solutions become quite unlikely.
> > > >
> > > > Is that true? I would think that dimensionality isn't the only relevant quantity, but also the number of centroids, yes? And if so, isn't the breaking point when the number of centroids > the ambient dimensionality? At that point, placing one centroid on another centroid becomes (equally) optimal.
> > > >
> > > > > To demonstrate, we examined mean and maximum absolute cosine similarity between optimized centroids, as a function of dimensionality and undercompleteness.
> > > >
> > > > To make sure I understand, what does undercompleteness mean? And what is a good null distribution to compare against for determining whether the optimized centroids overlap significantly (i.e. deviate from orthogonal)?

---

> > ### Comment · Reviewer_moiT · 2023-08-12
> > **Reviewer moiT response to Author Rebuttal**
> >
> > Hi all! Thank you for responding to my review! I appreciate you answering my questions. The clarifications make sense and I appreciate your disentangling the effect of the momentum encoder.
> >
> > Whether MMCR is significantly/sufficiently different from other SSL methods (e.g. TiCo, Wang et al. 2020's alignment+uniformity) still isn't exactly clear to me. The statement "The most important property of our objective is that it incentivizes global high dimensionality without relying on expectations over large numbers of pairwise comparisons" seems dubious since I would guess that one could rewrite those other loss functions by reorganizing some sums. For instance, in TiCo, instead of pairwise comparisons, one could first compute means, then perform pairwise dot products of means. I haven't thought this through but I no longer think detail this matters, as I explain below.
> >
> > My thinking about the review process also somewhat changed during ICML. I'm now of the opinion that most reviewers (perhaps myself included) are too high variance to be relied upon, and that perhaps a better process is just for reviewers to decide whether a paper is sufficiently well put together to be shared with others, so that the community can then sort out the paper's importance. With my new view, I think that this paper is definitely worth sharing and so I recommend that it should be accepted.
> >
> > To me, the value of this paper is showing that previous geometric manifold capacity analysis methods can be converted to a practical SSL method, plus many other contributions: theoretical result for the algorithm, comparison with biological data, empirical study of how the algorithm differs from other SSL algorithms, eigenspectrum decay.
> >
> > I've increased my score.

---

> > > ### Author Response · Authors · 2023-08-14
> > > **Clarifications on Toy Simulations**
> > >
> > > We are happy to clarify:
> > > - *"It seems odd to say that sETF solutions are optimal..."*: Point taken: we will describe the non-uniqueness of the optima in the revision.
> > >
> > > - *"Is that true? I would think that dimensionality..."*: Both the dimensionality and the number of centroids are relevant, and you are correct that when the number of centroids is larger than the dimensionality of the space replicating centroids can be an equally valid solution to the optimization problem. However the geometry of high dimensional spaces is such that randomly selected directions are nearly orthogonal to each other with high probability. Because of this as you move into higher dimensional spaces it becomes more likely that gradient descent chooses a solution that consists of approximately orthogonal centroids than one that replicates centroids. This stance is supported by the simulations above where we show that in a 512-D space, optimizing 4096 centroids to maximize the nuclear norm leads to a solution where no pair of centroids is very similar to each other.
> > >
> > > - *"To make sure I understand,..."*: By undercomplete we meant the case where there are more centroids than dimensions, we apologize for the ambiguity. In terms of a good null distribution for comparison, a reasonable choice would be the uniform distribution on the unit sphere. Randomly sampling 4096 512-D unit norm vectors, yields a distribution of (absolute) pairwise similarities with a maximum of ~0.2. From this we can conclude that optimizing the nuclear norm in this setting does not lead to significant overlap of centroid vectors any more than one would expect from a uniform distribution.

---

> > > ### Author Response · Authors · 2023-08-14
> > >
> > > Thank you for considering the contributions of our work, we are in complete agreement with your assessment (which is closely aligned with our summary in lines 40-50 of the submission).

---

### Author Rebuttal · Authors · 2023-08-10

We would like to thank the reviewers for their careful consideration of our submission. Their feedback has helped to identify several simple ways to improve our paper. Below we summarize some key revisions to be made in the case our work is accepted:

- The framing/introduction of our objective will be clarified. In particular we will change Equation (2) to only include the first term, as we immediately drop the second term in the following experiments.

- Relatedly, we will better motivate the use of “implicit” compression by noting the practical advantage (in terms of runtime).

- We will include a more thorough evaluation on object detection performance on the Pascal VOC dataset in the main text, to demonstrate that though our method is inspired by a theoretical analysis of linear classification the learned representations perform well on more complex computer vision tasks.

- We will better characterize the contribution of our neural data experiments, which show that different SSL methods trained in matched settings produce representations with diverse spectral/geometric properties but surprisingly similar neural predictivity.

---

> ### Author Response · Authors · 2023-08-14
> **Summary of Contributions**
>
> Large scale neural data are becoming widely available, presenting a challenge for their analysis and interpretation. Approaches focused on the geometry of neural activities have emerged as promising tools [1-4]. Among them, Manifold Capacity Theory (MCT), which analytically connects the geometry of neural manifolds to readout capacity, has been used to characterize the internal representations of ANN models of the brain, as well as biological neural data. However, it was unclear whether the MCT framework could serve as an effective tool for generating new ANN models for the brain. After all, how do we know if certain geometric measures are relevant for neural computation, unless they can generate a good model of the brain?
>
> Here we utilize the MCT (geometrical) framework, previously developed for characterizing representations in models and brains, to design an objective function that directly measures this population level geometric quantity. We demonstrate that:
> - This MCT objective is fundamentally different from nearly all previous SSL models, which rely on comparisons between pairs of points.
> - An ANN trained on this objective has performance close to SOTA levels in a variety of tasks
> - This trained ANN also predicts neural data well
> - MCT can also be used to characterize the internal representations of SSL models, suggesting a promising future direction on developing interpretable model-data comparison framework based on these measures.
>
> [1] Neural population geometry: An approach for understanding biological and artificial neural networks, Chung and Abbott, Current Opinion in Neurobiology 2021
>
> [2] A unifying perspective on neural manifolds and circuits for cognition, Langdon, Genkin, and Engel, Nature Reviews Neuroscience 2023
>
> [3] Neural tuning and representational geometry, Kriegeskorte and Wei, Nature Reviews Neuroscience, 2021
>
> [4] Neural representational geometry underlies few-shot concept learning, Sorscher, Ganguli, and Sompolinsky, PNAS 2022

---

### Decision · Program_Chairs · 2023-09-21

**Decision:**

Accept (poster)

**Comment:**

The paper presents a practical SSL algorithm which optimizes manifold capacity. The resulting model can perform well on linear-separable classification and are competitive with recognition and neural predictivity benchmarks. The approach is interesting and novel, well presented and justified, and well validated through evaluation and analysis. Furthermore, the approach to representation learning is interesting and stands out in comparison to popular SSL methods. I therefore recommend the paper to be accepted as a poster to NeurIPS.